# DIFFUSION-GAN: TRAINING GANs WITH DIFFUSION

**Zhendong Wang[1,2], Huangjie Zheng[1,2], Pengcheng He[2], Weizhu Chen[2], Mingyuan Zhou[1]**
[1]The University of Texas at Austin, [2]Microsoft Azure AI
{*zhendong.wang, huangjie.zheng*}@*utexas.edu*, {*penhe,wzchen*}@*microsoft.com*
*mingyuan.zhou@mccombs.utexas.edu*

## ABSTRACT

Generative adversarial networks (GANs) are challenging to train stably, and a promising remedy of injecting instance noise into the discriminator input has not been very effective in practice. In this paper, we propose Diffusion-GAN, a novel GAN framework that leverages a forward diffusion chain to generate Gaussian-mixture distributed instance noise. Diffusion-GAN consists of three components, including an adaptive diffusion process, a diffusion timestep-dependent discriminator, and a generator. Both the observed and generated data are diffused by the same adaptive diffusion process. At each diffusion timestep, there is a different noise-to-data ratio and the timestep-dependent discriminator learns to distinguish the diffused real data from the diffused generated data. The generator learns from the discriminator's feedback by backpropagating through the forward diffusion chain, whose length is adaptively adjusted to balance the noise and data levels. We theoretically show that the discriminator's timestep-dependent strategy gives consistent and helpful guidance to the generator, enabling it to match the true data distribution. We demonstrate the advantages of Diffusion-GAN over strong GAN baselines on various datasets, showing that it can produce more realistic images with higher stability and data efficiency than state-of-the-art GANs.

## 1 INTRODUCTION

Generative adversarial networks (GANs) (Goodfellow et al., 2014) and their variants (Brock et al., 2018; Karras et al., 2019; 2020a; Zhao et al., 2020) have achieved great success in synthesizing photo-realistic high-resolution images. GANs in practice, however, are known to suffer from a variety of issues ranging from non-convergence and training instability to mode collapse (Arjovsky and Bottou, 2017; Mescheder et al., 2018). As a result, a wide array of analyses and modifications has been proposed for GANs, including improving the network architectures (Karras et al., 2019; Radford et al., 2016; Sauer et al., 2021; Zhang et al., 2019), gaining theoretical understanding of GAN training (Arjovsky and Bottou, 2017; Heusel et al., 2017; Mescheder et al., 2017; 2018), changing the objective functions (Arjovsky et al., 2017; Bellemare et al., 2017; Deshpande et al., 2018; Li et al., 2017a; Nowozin et al., 2016; Zheng and Zhou, 2021; Yang et al., 2021), regularizing the weights and/or gradients (Arjovsky et al., 2017; Fedus et al., 2018; Mescheder et al., 2018; Miyato et al., 2018a; Roth et al., 2017; Salimans et al., 2016), utilizing side information (Wang et al., 2018; Zhang et al., 2017; 2020b), adding a mapping from the data to latent representation (Donahue et al., 2016; Dumoulin et al., 2016; Li et al., 2017b), and applying differentiable data augmentation (Karras et al., 2020a; Zhang et al., 2020a; Zhao et al., 2020).

A simple technique to stabilize GAN training is to inject instance noise, *i.e.*, to add noise to the discriminator input, which can widen the support of both the generator and discriminator distributions and prevent the discriminator from overfitting (Arjovsky and Bottou, 2017; Sønderby et al., 2017). However, this technique is hard to implement in practice, as finding a suitable noise distribution is challenging (Arjovsky and Bottou, 2017). Roth et al. (2017) show that adding instance noise to the high-dimensional discriminator input does not work well, and propose to approximate it by adding a zero-centered gradient penalty on the discriminator. This approach is theoretically and empirically shown to converge in Mescheder et al. (2018), who also demonstrate that adding zero-centered gradient penalties to non-saturating GANs can result in stable training and better or comparable generation quality compared to WGAN-GP (Arjovsky et al., 2017). However, Brock

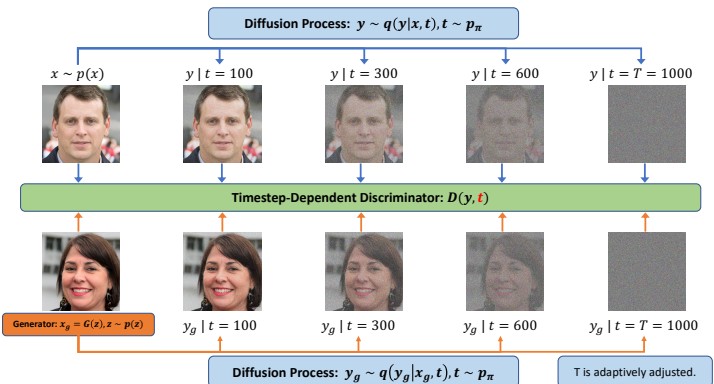

Figure 1: Flowchart for Diffusion-GAN. The top-row images represent the forward diffusion process of a real image, while the bottom-row images represent the forward diffusion process of a generated fake image. The discriminator learns to distinguish a diffused real image from a diffused fake image at all diffusion steps.

et al. (2018) caution that zero-centered gradient penalties and other similar regularization methods may stabilize training at the cost of generation performance. To the best of our knowledge, there has been no existing work that is able to empirically demonstrate the success of using instance noise in GAN training on high-dimensional image data.

To inject proper instance noise that can facilitate GAN training, we introduce Diffusion-GAN, which uses a diffusion process to generate Gaussian-mixture distributed instance noise. We show a graphical representation of Diffusion-GAN in Figure 1. In Diffusion-GAN, the input to the diffusion process is either a real or a generated image, and the diffusion process consists of a series of steps that gradually add noise to the image. The number of diffusion steps is not fixed, but depends on the data and the generator. We also design the diffusion process to be differentiable, which means that we can compute the derivative of the output with respect to the input. This allows us to propagate the gradient from the discriminator to the generator through the diffusion process, and update the generator accordingly. Unlike vanilla GANs, which compare the real and generated images directly, Diffusion-GAN compares the noisy versions of them, which are obtained by sampling from the Gaussian mixture distribution over the diffusion steps, with the help of our timestep-dependent discriminator. This distribution has the property that its components have different noise-to-data ratios, which means that some components add more noise than others. By sampling from this distribution, we can achieve two benefits: first, we can stabilize the training by easing the problem of vanishing gradient, which occurs when the data and generator distributions are too different; second, we can augment the data by creating different noisy versions of the same image, which can improve the data efficiency and the diversity of the generator. We provide a theoretical analysis to support our method, and show that the min-max objective function of Diffusion-GAN, which measures the difference between the data and generator distributions, is continuous and differentiable everywhere. This means that the generator in theory can always receive a useful gradient from the discriminator, and improve its performance.

Our main contributions include: 1) We show both theoretically and empirically how the diffusion process can be utilized to provide a model- and domain-agnostic differentiable augmentation, enabling data-efficient and leaking-free stable GAN training. 2) Extensive experiments show that Diffusion-GAN boosts the stability and generation performance of strong baselines, including Style-GAN2 (Karras et al., 2020b), Projected GAN (Sauer et al., 2021), and InsGen (Yang et al., 2021), achieving state-of-the-art results in synthesizing photo-realistic images, as measured by both the Fréchet Inception Distance (FID) (Heusel et al., 2017) and Recall score (Kynkäänniemi et al., 2019).

## 2 PRELIMINARIES: GANs AND DIFFUSION-BASED GENERATIVE MODELS

GANs (Goodfellow et al., 2014) are a class of generative models that aim to learn the data distribution $p(x)$ of a target dataset by setting up a min-max game between two neural networks: a generator and a discriminator. The generator $G$ takes as input a random noise vector $z$ sampled from a simple prior distribution $p(z)$, such as a standard normal or uniform distribution, and tries to produce realistic-looking samples $G(z)$ that resemble the data. The discriminator $D$ receives either

a real data sample $\boldsymbol{x}$ drawn from $p(\boldsymbol{x})$ or a fake sample $G(\boldsymbol{z})$ generated by $G$, and tries to correctly classify them as real or fake. The goal of $G$ is to fool $D$ into making mistakes, while the goal of $D$ is to accurately distinguish $G(\boldsymbol{z})$ from $\boldsymbol{x}$. The min-max objective function of GANs is given by

$$\min_G \max_D V(G, D) = \mathbb{E}_{\boldsymbol{x} \sim p(\boldsymbol{x})}[\log(D(\boldsymbol{x}))] + \mathbb{E}_{\boldsymbol{z} \sim p(\boldsymbol{z})}[\log(1 - D(G(\boldsymbol{z})))].$$

In practice, this vanilla objective function is often modified to improve the stability and performance of GANs(Goodfellow et al., 2014; Miyato et al., 2018a; Fedus et al., 2018), but the general idea of adversarial learning between $G$ and $D$ remains the same.

Diffusion-based generative models (Ho et al., 2020b; Sohl-Dickstein et al., 2015; Song and Ermon, 2019) assume $p_\theta(\boldsymbol{x}_0) := \int p_\theta(\boldsymbol{x}_{0:T}) d\boldsymbol{x}_{1:T}$, where $\boldsymbol{x}_1, \ldots, \boldsymbol{x}_T$ are latent variables of the same dimensionality as the data $\boldsymbol{x}_0 \sim p(\boldsymbol{x}_0)$. There is a forward diffusion chain that gradually adds noise to the data $\boldsymbol{x}_0 \sim q(\boldsymbol{x}_0)$ in $T$ steps with pre-defined variance schedule $\beta_t$ and variance $\sigma^2$:

$$q(\boldsymbol{x}_{1:T} \,|\, \boldsymbol{x}_0) := \prod_{t=1}^T q(\boldsymbol{x}_t \,|\, \boldsymbol{x}_{t-1}), \quad q(\boldsymbol{x}_t \,|\, \boldsymbol{x}_{t-1}) := \mathcal{N}(\boldsymbol{x}_t; \sqrt{1 - \beta_t}\boldsymbol{x}_{t-1}, \beta_t \sigma^2 \boldsymbol{I}).$$

A notable property is that $\boldsymbol{x}_t$ at an arbitrary time-step $t$ can be sampled in closed form as

$$q(\boldsymbol{x}_t \,|\, \boldsymbol{x}_0) = \mathcal{N}(\boldsymbol{x}_t; \sqrt{\bar{\alpha}_t}\boldsymbol{x}_0, (1 - \bar{\alpha}_t)\sigma^2 \boldsymbol{I}), \quad \text{where } \alpha_t := 1 - \beta_t, \ \bar{\alpha}_t := \prod_{s=1}^t \alpha_s. \tag{1}$$

A variational lower bound (Blei et al., 2017) is then used to optimize the reverse diffusion chain as

$$p_\theta(\boldsymbol{x}_{0:T}) := \mathcal{N}(\boldsymbol{x}_T; \boldsymbol{0}, \sigma^2 \boldsymbol{I}) \prod_{t=1}^T p_\theta(\boldsymbol{x}_{t-1} \,|\, \boldsymbol{x}_t).$$

# 3 DIFFUSION-GAN: METHOD AND THEORETICAL ANALYSIS

To construct Diffusion-GAN, we describe how to inject instance noise via diffusion, how to train the generator by backpropagating through the forward diffusion process, and how to adaptively adjust the diffusion intensity. We further provide theoretical analysis illustrated with a toy example.

## 3.1 INSTANCE NOISE INJECTION VIA DIFFUSION

We aim to generate realistic samples $\boldsymbol{x}_g$ from a generator network $G$ that maps a latent variable $\boldsymbol{z}$ sampled from a simple prior distribution $p(\boldsymbol{z})$ to a high-dimensional data space, such as images. The distribution of generator samples $\boldsymbol{x}_g = G(\boldsymbol{z})$, $\boldsymbol{z} \sim p(\boldsymbol{z})$ is denoted by $p_g(\boldsymbol{x}) = \int p(\boldsymbol{x}_g \,|\, \boldsymbol{z})p(\boldsymbol{z})d\boldsymbol{z}$. To make the generator more robust and diverse, we inject instance noise into the generated samples $\boldsymbol{x}_g$ by applying a diffusion process that adds Gaussian noise at each step. The diffusion process can be seen as a Markov chain that starts from the original sample $\boldsymbol{x}$ and gradually erases its information until reaching a noise level $\sigma^2$ after $T$ steps.

We define a mixture distribution $q(\boldsymbol{y} \,|\, \boldsymbol{x})$ that models the noisy samples $\boldsymbol{y}$ obtained at any step of the diffusion process, with a mixture weight $\pi_t$ for each step $t$. The mixture components $q(\boldsymbol{y} \,|\, \boldsymbol{x}, t)$ are Gaussian distributions with mean proportional to $\boldsymbol{x}$ and variance depending on the noise level at step $t$. We use the same diffusion process and mixture distribution for both the real samples $\boldsymbol{x} \sim p(\boldsymbol{x})$ and the generated samples $\boldsymbol{x}_g \sim p_g(\boldsymbol{x})$. More specifically, the diffusion-induced mixture distributions are expressed as

$$\boldsymbol{x} \sim p(\boldsymbol{x}), \, \boldsymbol{y} \sim q(\boldsymbol{y} \,|\, \boldsymbol{x}), \ \ q(\boldsymbol{y} \,|\, \boldsymbol{x}) := \textstyle\sum_{t=1}^T \pi_t q(\boldsymbol{y} \,|\, \boldsymbol{x}, t),$$

$$\boldsymbol{x}_g \sim p_g(\boldsymbol{x}), \, \boldsymbol{y}_g \sim q(\boldsymbol{y}_g \,|\, \boldsymbol{x}_g), \ \ q(\boldsymbol{y}_g \,|\, \boldsymbol{x}_g) := \textstyle\sum_{t=1}^T \pi_t q(\boldsymbol{y}_g \,|\, \boldsymbol{x}_g, t),$$

where $q(\boldsymbol{y} \,|\, \boldsymbol{x})$ is a $T$-component mixture distribution, the mixture weights $\pi_t$ are non-negative and sum to one, and the mixture components $q(\boldsymbol{y} \,|\, \boldsymbol{x}, t)$ are obtained via diffusion as in Equation (1), expressed as

$$q(\boldsymbol{y} \,|\, \boldsymbol{x}, t) = \mathcal{N}(\boldsymbol{y}; \sqrt{\bar{\alpha}_t}\boldsymbol{x}, (1 - \bar{\alpha}_t)\sigma^2 \boldsymbol{I}). \tag{2}$$

Samples from this mixture can be drawn as $t \sim p_\pi := \text{Discrete}(\pi_1, \ldots, \pi_T), \ \boldsymbol{y} \sim q(\boldsymbol{y} \,|\, \boldsymbol{x}, t)$.

By sampling $\boldsymbol{y}$ from this mixture distribution, we can obtain noisy versions of both real and generated samples with varying degrees of noise. The more steps we take in the diffusion process, the more noise we add to $\boldsymbol{y}$ and the less information we preserve from $\boldsymbol{x}$. We can then use this diffusion-induced mixture distribution to train a timestep-dependent discriminator $D$ that distinguishes between real and generated noisy samples, and a generator $G$ that matches the distribution of generated noisy samples to the distribution of real noisy samples. Next we introduce Diffusion-GAN that trains its discriminator and generator with the help of the diffusion-induced mixture distribution.

## 3.2 ADVERSARIAL TRAINING

The Diffusion-GAN trains its generator and discriminator by solving a min-max game objective as

$$V(G, D) = \mathbb{E}_{\boldsymbol{x} \sim p(\boldsymbol{x}), t \sim p_\pi, \boldsymbol{y} \sim q(\boldsymbol{y} \,|\, \boldsymbol{x}, t)}[\log(D_\phi(\boldsymbol{y}, t))] + \mathbb{E}_{\boldsymbol{z} \sim p(\boldsymbol{z}), t \sim p_\pi, \boldsymbol{y}_g \sim q(\boldsymbol{y} \,|\, G_\theta(\boldsymbol{z}), t)}[\log(1 - D_\phi(\boldsymbol{y}_g, t))]. \quad (3)$$

Here, $p(\boldsymbol{x})$ is the true data distribution, $p_\pi$ is a discrete distribution that assigns different weights $\pi_t$ to each diffusion step $t \in \{1, \ldots, T\}$, and $q(\boldsymbol{y} \,|\, \boldsymbol{x}, t)$ is the conditional distribution of the perturbed sample $\boldsymbol{y}$ given the original data $\boldsymbol{x}$ and the diffusion step $t$. By Equation (2), with Gaussian reparameterization, the perturbation function could be written as $\boldsymbol{y} = \sqrt{\bar{\alpha}_t}\boldsymbol{x} + \sqrt{1 - \bar{\alpha}_t}\sigma\boldsymbol{\epsilon}$, where $1 - \bar{\alpha}_t = 1 - \prod_{s=1}^t \alpha_s$ is the cumulative noise level at step $t$, $\sigma$ is a scale factor, and $\boldsymbol{\epsilon} \sim \mathcal{N}(0, \boldsymbol{I})$ is a Gaussian noise.

The objective function in Equation (3) encourages the discriminator to assign high probabilities to the perturbed real data and low probabilities to the perturbed generated data, for any diffusion step $t$. The generator, on the other hand, tries to produce samples that can deceive the discriminator at any diffusion step $t$. Note that the perturbed generated sample $\boldsymbol{y}_g \sim q(\boldsymbol{y} \,|\, G_\theta(\boldsymbol{z}), t)$ can be rewritten as $\boldsymbol{y}_g = \sqrt{\bar{\alpha}_t}G_\theta(\boldsymbol{z}) + \sqrt{(1 - \bar{\alpha}_t)}\sigma\boldsymbol{\epsilon}, \boldsymbol{\epsilon} \sim \mathcal{N}(0, \boldsymbol{I})$. This means that the objective function in Equation (3) is differentiable with respect to the generator parameters, and we can use gradient descent to optimize it with back-propagation.

The objective function Equation (3) is similar to the one used by the original GAN (Goodfellow et al., 2014), except that it involves the diffusion steps and the perturbation functions. We can show that this objective function also minimizes an approximation of the *Jensen–Shannon (JS) divergence* between the true and the generated distributions, but with respect to the perturbed samples and the diffusion steps, as follows:

$$\mathcal{D}_{\text{JS}}(p(\boldsymbol{y}, t) || p_g(\boldsymbol{y}, t)) = \mathbb{E}_{t \sim p_\pi}[\mathcal{D}_{\text{JS}}(p(\boldsymbol{y} \,|\, t) || p_g(\boldsymbol{y} \,|\, t))]. \quad (4)$$

The JS divergence measures the dissimilarity between two probability distributions, and it reaches its minimum value of zero when the two distributions are identical. The proof of the equality in Equation (4) is given in Appendix C. A natural question that arises from this result is whether minimizing the JS divergence between the perturbed distributions implies minimizing the JS divergence between the original distributions, *i.e.*, whether the optimal generator for Equation (3) is also the optimal generator for $\mathcal{D}_{\text{JS}}(p(\boldsymbol{x}) || p_g(\boldsymbol{x}))$. We will answer this question affirmatively and provide a theoretical justification in Section 3.4.

## 3.3 ADAPTIVE DIFFUSION

With the help of the perturbation function and timestep dependency, we have a new strategy to optimize the discriminator. We want the discriminator $D$ to have a challenging task, neither too easy to allow overfitting the data (Karras et al., 2020a; Zhao et al., 2020) nor too hard to impede learning. Therefore, we adjust the intensity of the diffusion process, which adds noise to both $\boldsymbol{y}$ and $\boldsymbol{y}_g$, depending on how much $D$ can distinguish them. When the diffusion step $t$ is larger, the noise-to-data ratios are higher and the task is harder. We use $1 - \bar{\alpha}_t$ to measure the intensity of the diffusion, which increases as $t$ grows. To control the diffusion intensity, we adaptively modify the maximum number of steps $T$.

Our strategy is to make the discriminator learn from the easiest samples first, which are the original data samples, and then gradually increase the difficulty by feeding it samples from larger $t$. To do this, we use a self-paced schedule for $T$, which depends on a metric $r_d$ that estimates how much the discriminator overfits to the data:

$$r_d = \mathbb{E}_{\boldsymbol{y}, t \sim p(\boldsymbol{y}, t)}[\text{sign}(D_\phi(\boldsymbol{y}, t) - 0.5)], \quad T = T + \text{sign}(r_d - d_{target}) * C, \quad (5)$$

where $r_d$ is the same as in Karras et al. (2020a) and $C$ is a constant. We calculate $r_d$ and update $T$ every four minibatches. We have two options for the distribution $p_\pi$ that we use to sample $t$ for the diffusion process:

$$t \sim p_\pi := \begin{cases} \text{uniform:} & \text{Discrete}\left(\frac{1}{T}, \frac{1}{T}, \ldots, \frac{1}{T}\right), \\ \text{priority:} & \text{Discrete}\left(\frac{1}{\sum_{t=1}^T t}, \frac{2}{\sum_{t=1}^T t}, \ldots, \frac{T}{\sum_{t=1}^T t}\right), \end{cases} \quad (6)$$

The 'priority' option gives more weight to larger $t$, which means the discriminator will see more new samples from the new steps when $T$ increases. This is because we want the discriminator to focus on

the new and harder samples that it has not seen before, as this indicates that it is confident about the easier ones. Note that even with the 'priority' option, the discriminator can still see samples from smaller $t$, because $q(\boldsymbol{y} \mid \boldsymbol{x})$ is a mixture of Gaussians that covers all steps of the diffusion chain.

To avoid sudden changes in $T$ during training, we use an exploration list $\boldsymbol{t}_{epl}$ that contains $t$ values sampled from $p_\pi$. We keep $\boldsymbol{t}_{epl}$ fixed until we update $T$, and we sample $t$ from $\boldsymbol{t}_{epl}$ to generate noisy samples for the discriminator. This way, the model can explore each $t$ sufficiently before moving to a higher $T$. We give the details of training Diffusion-GAN in Algorithm 1 in Appendix F.

### 3.4 THEORETICAL ANALYSIS WITH EXAMPLES

To better understand the theoretical properties of our proposed method, we present two theorems that address two important questions about the use of diffusion-based instance noise injection for training GANs. The proofs of these theorems are deferred to Appendix B. The first question, denoted as **(a)**, is whether adding noise to the real and generated samples in a diffusion process can facilitate the learning. The second question, denoted as **(b)**, is whether minimizing the JS divergence between the joint distributions of the noisy samples and the noise levels, $p(\boldsymbol{y}, t)$ and $p_g(\boldsymbol{y}, t)$, can lead to the same optimal generator as minimizing the JS divergence between the original distributions of the real and generated samples, $p(\boldsymbol{x})$ and $p_g(\boldsymbol{x})$.

To answer **(a)**, we prove that for any choice of noise level $t$ and any choice of convex function $f$, the $f$-divergence (Nowozin et al., 2016) between the marginal distributions of the noisy real and generated samples, $q(\boldsymbol{y} \mid t)$ and $q(\boldsymbol{y}_g \mid t)$, is a smooth function that can be computed and optimized by the discriminator. This implies that the diffusion-based noise injection does not introduce any singularity or discontinuity in the objective function of the GAN. The JS divergence is a special case of $f$-divergence, where $f(u) = -\log(2u) - \log(2 - 2u)$.

**Theorem 1** (Valid gradients anywhere for GANs training). *Let $p(\boldsymbol{x})$ be a fixed distribution over $\mathcal{X}$ and $\boldsymbol{z}$ be a random noise over another space $\mathcal{Z}$. Denote $G_\theta : \mathcal{Z} \to \mathcal{X}$ as a function with parameter $\theta$ and input $\boldsymbol{z}$ and $p_g(\boldsymbol{x})$ as the distribution of $G_\theta(\boldsymbol{z})$. Let $q(\boldsymbol{y} \mid \boldsymbol{x}, t) = \mathcal{N}(\boldsymbol{y}; \sqrt{\bar{\alpha}_t}\boldsymbol{x}, (1 - \bar{\alpha}_t)\sigma^2\boldsymbol{I})$, where $\bar{\alpha}_t \in (0, 1)$ and $\sigma > 0$. Let $q(\boldsymbol{y} \mid t) = \int p(\boldsymbol{x})q(\boldsymbol{y} \mid \boldsymbol{x}, t)d\boldsymbol{x}$ and $q_g(\boldsymbol{y} \mid t) = \int p_g(\boldsymbol{x})q(\boldsymbol{y} \mid \boldsymbol{x}, t)d\boldsymbol{x}$. Then, $\forall t$, if function $G_\theta$ is continuous and differentiable, the f-divergence $\mathcal{D}_f(q(\boldsymbol{y} \mid t)||q_g(\boldsymbol{y} \mid t))$ is continuous and differentiable with respect to $\theta$.*

Theorem 1 shows that with the help of diffusion noise injection by $q(\boldsymbol{y} \mid \boldsymbol{x}, t)$, $\forall t$, $\boldsymbol{y}$ and $\boldsymbol{y}_g$ are defined on the same support space, the whole $\mathcal{X}$, and $\mathcal{D}_f(q(\boldsymbol{y} \mid t)||q_g(\boldsymbol{y} \mid t))$ is continuous and differentiable everywhere. Then, one natural question is what if $\mathcal{D}_f(q(\boldsymbol{y} \mid t)||q_g(\boldsymbol{y} \mid t))$ keeps a near constant value and hence provides little useful gradient. Hence, we empirically show that by injecting noise through a mixture defined over all steps of the diffusion chain, there is always a good chance that a sufficiently large $t$ is sampled to provide a useful gradient, via the toy example below.

**Toy example.** We use the same simple example from Arjovsky et al. (2017) to illustrate our method. Let $\boldsymbol{x} = (0, z)$ be the real data and $\boldsymbol{x}_g = (\theta, z)$ be the data generated by a one-parameter generator, where $z$ is a uniform random variable in $[0, 1]$. The JS divergence between the real and the generated distributions, $\mathcal{D}_{JS}(p(\boldsymbol{x})||p(\boldsymbol{x}_g))$, is discontinuous: it is $\log 2$ when $\theta = 0$ and zero otherwise, so it does not provide a useful gradient to guide $\theta$ towards zero.

We introduce diffusion-based noise to both the real and the generated data, as shown in the first row of Figure 2. The noisy data, $\boldsymbol{y}$ and $\boldsymbol{y}_g$, have supports that cover the whole space $\mathbb{R}^2$ and their densities overlap more or less depending on the diffusion step $t$. In the second row, left, of Figure 2, we plot how the JS divergence between the noisy distributions, $\mathcal{D}_{JS}(q(\boldsymbol{y} \mid t)||q_g(\boldsymbol{y} \mid t))$, varies with $\theta$ for different $t$ values. The black line with $t = 0$ is the original JS divergence, which has a discontinuity at $\theta = 0$. As $t$ increases, the JS divergence curves become smoother and have non-zero gradients for a larger range of $\theta$. However, some values of $t$, such as $t = 200$ in this example, still have flat regions where the JS divergence is nearly constant. To avoid this, we use a mixture of all steps to ensure that there is always a high chance of getting informative gradients.

For the discriminator optimization, as shown in the second row, right, of Figure 2, the optimal discriminator under the original JS divergence is discontinuous and unattainable. With diffusion-based noise, the optimal discriminator changes with $t$: a smaller $t$ makes it more confident and a larger $t$ makes it more cautious. Thus the diffusion acts like a scale to balance the power of the

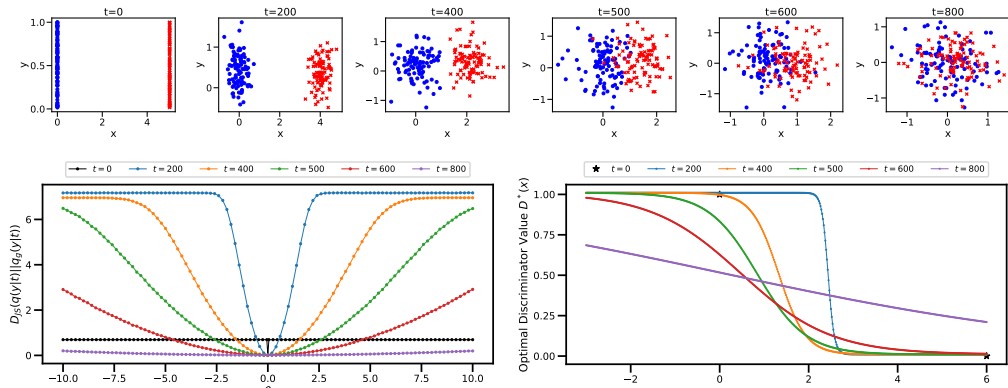

Figure 2: The toy example inherited from Arjovsky et al. (2017). The first row plots the distributions of data with diffusion noise injected for $t$. The second row shows the JS divergence and the optimal discriminator value with and without our noise injection.

discriminator. This suggests the use of a differentiable forward diffusion chain that can provide various levels of gradient smoothness to help the generator training.

**Theorem 2** (Non-leaking noise injection). *Let $\boldsymbol{x} \sim p(\boldsymbol{x}), \boldsymbol{y} \sim q(\boldsymbol{y} \,|\, \boldsymbol{x})$ and $\boldsymbol{x}_g \sim p_g(\boldsymbol{x}), \boldsymbol{y}_g \sim q(\boldsymbol{y}_g \,|\, \boldsymbol{x}_g)$, where $q(\boldsymbol{y} \,|\, \boldsymbol{x})$ is the transition density. Given certain $q(\boldsymbol{y} \,|\, \boldsymbol{x})$, if $\boldsymbol{y}$ could be reparameterized into $\boldsymbol{y} = f(\boldsymbol{x}) + h(\boldsymbol{\epsilon})$, $\boldsymbol{\epsilon} \sim p(\boldsymbol{\epsilon})$, where $p(\boldsymbol{\epsilon})$ is a known distribution, and both $f$ and $h$ are one-to-one mapping functions, then we could have $p(\boldsymbol{y}) = p_g(\boldsymbol{y}) \Leftrightarrow p(\boldsymbol{x}) = p_g(\boldsymbol{x})$.*

To answer question **(b)**, we present Theorem 2, which shows a sufficient condition for the equality of the original and the augmented data distributions. By Theorem 2, the function $f$ maps each $\boldsymbol{x}$ to a unique $\boldsymbol{y}$, the function $h$ maps each $\boldsymbol{\epsilon}$ to a unique noise term, and the distribution of $\boldsymbol{\epsilon}$ is known and independent of $\boldsymbol{x}$. Under these assumptions, the theorem proves that the distribution of $\boldsymbol{y}$ is the same as the distribution of $\boldsymbol{y}_g$, if and only if the distribution of $\boldsymbol{x}$ is the same as the distribution of $\boldsymbol{x}_g$. If we take $\boldsymbol{y} \,|\, t$ as the $\boldsymbol{y}$ introduced in the theorem, then for $\forall t$, Equation (2) fits the assumption made. This means that, by minimizing the divergence between $q(\boldsymbol{y} \,|\, t)$ and $q_g(\boldsymbol{y} \,|\, t)$, which is the same as minimizing the divergence between $p(\boldsymbol{x}) \,|\, t$ and $p_g(\boldsymbol{x}) \,|\, t$, we are also minimizing the divergence between $p(\boldsymbol{x})$ and $p_g(\boldsymbol{x})$. This implies that the noise injection does not affect the quality of the generated samples, and we can safely use our noise injection to improve the training of the generative model.

### 3.5 RELATED WORK

The proposed Diffusion-GAN can be related to previous works on stabilizing the GAN training, building diffusion-based generative models, and constructing differential augmentation for data-efficient GAN training. A detailed discussion on these related works is deferred to Appendix A.

## 4 EXPERIMENTS

We conduct extensive experiments to answer the following questions: **(a)** Will Diffusion-GAN outperform state-of-the-art GAN baselines on benchmark datasets? **(b)** Will the diffusion-based noise injection help the learning of GANs in domain-agnostic tasks? **(c)** Will our method improve the performance of data-efficient GANs trained with a very limited amount of data?

**Datasets.** We conduct experiments on image datasets ranging from low-resolution (*e.g.*, $32 \times 32$) to high-resolution (*e.g.*, $1024 \times 1024$) and from low-diversity to high-diversity: CIFAR-10 (Krizhevsky, 2009), STL-10 (Coates et al., 2011), LSUN-Bedroom (Yu et al., 2015), LSUN-Church (Yu et al., 2015), AFHQ(Cat/Dog/Wild) (Choi et al., 2020), and FFHQ (Karras et al., 2019). More details on these benchmark datasets are provided in Appendix E.

**Evaluation protocol.** We measure image quality using FID (Heusel et al., 2017). Following Karras et al. (2019; 2020b), we measure FID using 50k generated samples, with the full training set used

Table 1: Image generation results on benchmark datasets: CIFAR-10, CelebA, STL-10, LSUN-Bedroom, LSUN-Church, and FFHQ. We highlight the best and second best results in each column with **bold** and underline, respectively. Lower FIDs indicate better fidelity, while higher Recalls indicate better diversity.

| Methods | CIFAR-10 $(32 \times 32)$ | | CelebA $(64 \times 64)$ | | STL-10 $(64 \times 64)$ | | LSUN-Bedroom $(256 \times 256)$ | | LSUN-Church $(256 \times 256)$ | | FFHQ $(1024 \times 1024)$ | |
|---|---|---|---|---|---|---|---|---|---|---|---|---|
| | FID | Recall | FID | Recall | FID | Recall | FID | Recall | FID | Recall | FID | Recall |
| StyleGAN2 (Karras et al., 2020a) | 8.32* | 0.41* | 2.32 | 0.55 | 11.70 | 0.44 | 3.98 | 0.32 | 3.93 | 0.39 | 4.41 | 0.42 |
| StyleGAN2 + DiffAug (Zhao et al., 2020) | 5.79* | 0.42* | 2.75 | 0.52 | 12.97 | 0.39 | 4.25 | 0.19 | 4.66 | 0.33 | 4.46 | 0.41 |
| StyleGAN2 + ADA (Karras et al., 2020a) | **2.92*** | 0.49* | 2.49 | 0.53 | 13.72 | 0.36 | 7.89 | 0.05 | 4.12 | 0.18 | 4.47 | 0.41 |
| Diffusion StyleGAN2 | 3.19 | **0.58** | **1.69** | **0.67** | **11.43** | **0.45** | **3.65** | **0.32** | **3.17** | **0.42** | **2.83** | **0.49** |

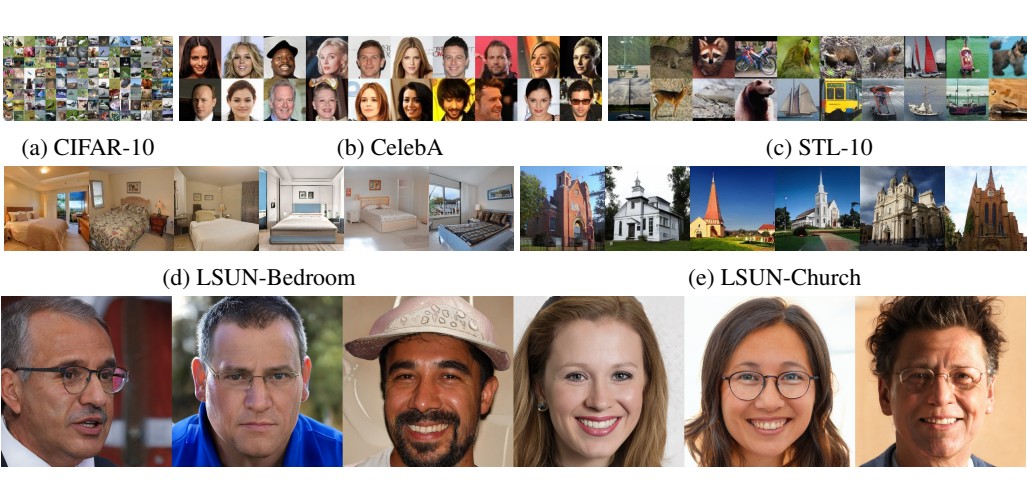

(a) CIFAR-10     (b) CelebA     (c) STL-10

(d) LSUN-Bedroom     (e) LSUN-Church

(f) FFHQ

Figure 3: Randomly generated images from Diffusion StyleGAN2 trained on CIFAR-10, CelebA, STL-10, LSUN-Bedroom, LSUN-Church, and FFHQ datasets.

as reference. We use the number of real images shown to the discriminator to evaluate convergence (Karras et al., 2020a; Sauer et al., 2021). Unless specified otherwise, all models are trained with 25 million images to ensure convergence (these trained with more or fewer images are specified in table captions). We further report the improved *Recall* score introduced by Kynkäänniemi et al. (2019) to measure the sample diversity of generative models.

**Implementations and resources.** We build Diffusion-GANs based on the code of StyleGAN2 (Karras et al., 2020b), ProjectedGAN (Sauer et al., 2021), and InsGen (Yang et al., 2021) to answer questions **(a)**, **(b)**, and **(c)**, respectively. Diffusion GANs inherit from their corresponding base GANs all their network architectures and training hyperparamters, whose details are provided in Appendix G. Specifically for StyleGAN2 and InsGen, we construct the discriminator as $D_\phi(\boldsymbol{y}, t)$, where $t$ is injected via their mapping network. For ProjectedGAN, we empirically find $t$ in the discriminator could be ignored to simplify the implementation and minimize the modifications to ProjectedGAN. More implementation details are provided in Appendix H. By applying our diffusion-based noise injection, we denote our models as Diffusion StyleGAN2/ProjectedGAN/InsGen. In the following experiments, we train related models with their official code if the results are unavailable, while others are all reported from references and marked with *. We run all our experiments with either 4 or 8 NVIDIA V100 GPUs depending on the demands of the inherited training configurations.

### 4.1 COMPARISON TO STATE-OF-THE-ART GANS

We compare Diffusion-GAN with its state-of-the-art GAN backbone, StyleGAN2 (Karras et al., 2020a), and to evaluate its effectiveness from the data augmentation perspective, we compare it with both StyleGAN2 + DiffAug (Zhao et al., 2020) and StyleGAN2 + ADA (Karras et al., 2020a), in terms of both sample fidelity (FID) and sample diversity (Recall) over extensive benchmark datasets.

We present the quantitative and qualitative results in Table 1 and Figure 3. Qualitatively, these generated images from Diffusion StyleGAN2 are all photo-realistic and have good diversity, ranging from low-resolution $(32 \times 32)$ to high-resolution $(1024 \times 1024)$. Additional randomly generated images

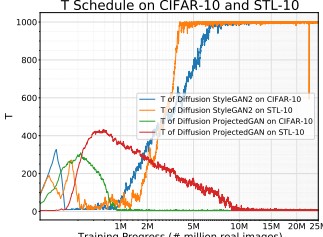 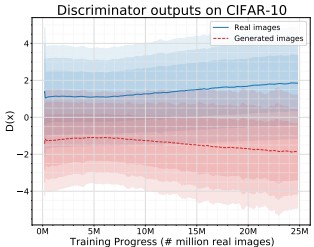

Figure 4: Plot of adaptively adjusted maximum diffusion steps $T$ and discriminator outputs of Diffusion-GANs.

can be found in Appendix L. Quantitatively, Diffusion StyleGAN2 outperforms all the GAN baselines in generation diversity, as measured by Recall, on all 6 benchmark datasets and outperforms them in FID by a clear margin on 5 out of the 6 benchmark datasets.

From the data augmentation perspective, we observe that Diffusion StyleGAN2 always clearly outperforms the backbone model StyleGAN2 across various datasets, which empirically validates our Theorem 2. By contrast, both the ADA (Karras et al., 2020b) and Diffaug (Zhao et al., 2020) techniques could sometimes impair the generation performance on sufficiently large datasets, *e.g.*, LSUN-Bedroom and LSUN-Church, which is also observed by Yang et al. (2021) on FFHQ. This is possibly because their risk of leaking augmentation overshadows the benefits of data augmentation.

To investigate how the adaptive diffusion process works during training, we illustrate in Figure 4 the convergence of the maximum timestep $T$ in our adaptive diffusion and discriminator outputs. We see that $T$ is adaptively adjusted: The $T$ for Diffusion StyleGAN2 increases as the training goes while the $T$ for Diffusion ProjectedGAN first goes up and then goes down. Note that the $T$ is adjusted according to the overfitting status of the discriminator. The second panel shows that trained with the diffusion-based mixture distribution, the discriminator is always well behaved and provides useful learning signals for the generator, which validates our analysis in Section 3.4 and Theorem 1.

**Memory and time costs.** Generally speaking, the memory and time costs of a Diffusion-GAN are comparable to those of the corresponding GAN baseline. More specifically, switching from ADA (Karras et al., 2020a) to our diffusion-based augmentation, the added memory cost is negative, the added training time cost is negative, and the added inference time cost is zero. For example, for CIFAR-10, with four NVIDIA V100 GPUs, the training time for each 4k images is around 8.0s for StyleGAN2, 9.8s for StyleGAN2-ADA, and 9.5s for Diffusion-StyleGAN2.

## 4.2 EFFECTIVENESS OF DIFFUSION-GAN FOR DOMAIN-AGNOSTIC AUGMENTATION

To verify whether our method is domain-agnostic, we apply Diffusion-GAN onto the input feature vectors of GANs. We conduct experiments on both low-dimensional and high-dimensional feature vectors, for which commonly used image augmentation methods are no longer applicable.

**25-Gaussians Example.** We conduct experiments on the popular 25-Gaussians generation task. The 25-Gaussians dataset is a 2-D toy data, generated by a mixture of 25 two-dimensional Gaussian distributions. Each data point is a 2-dimensional feature vector. We train a small GAN model, whose generator and discriminator are both parameterized by multilayer perceptrons (MLPs), with two 128-unit hidden layers and LeakyReLu nonlinearities.

The training results are shown in Figure 5. We observe that the vanilla GAN exhibits severe mode collapsing, capturing only a few modes. Its discriminator outputs of real and fake samples depart from each other very quickly. This implies a strong overfitting of the discriminator happened so that the discriminator stops providing useful learning signals for the generator. However, Diffusion-GAN successfully captures all the 25 Gaussian modes and the discriminator is under control to continuously provide useful learning signals. We interpret the improvement from two perspectives: First, non-leaking augmentation helps provide more information about the data space; Second, the discriminator is well behaved given the adaptively adjusted diffusion-based noise injection.

**ProjectedGAN.** To verify that our adaptive diffusion-based noise injection could benefit the learning of GANs on high-dimensional feature vectors, we directly apply it to the discriminator feature

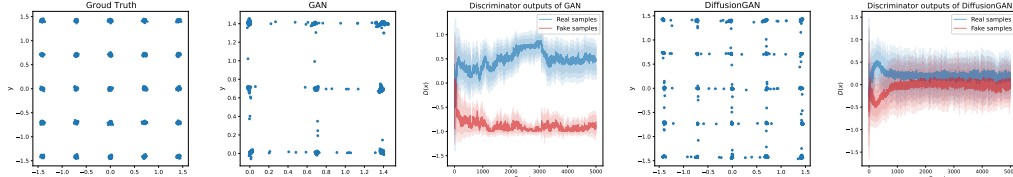

Figure 5: The 25-Gaussians example. We show the true data samples, the generated samples from vanilla GANs, the discriminator outputs of the vanilla GANs, the generated samples from our Diffusion-GAN, and the discriminator outputs of Diffusion-GAN.

space of ProjectedGAN (Sauer et al., 2021). ProjectedGANs generally leverage pre-trained neural networks to extract meaningful features for the adversarial learning of the discriminator and generator. Following Sauer et al. (2021), we adaptively diffuse the feature vectors extracted by EfficientNet-v0 and keep all the other training parts unchanged. We report the performance of Diffusion ProjectedGAN on several benchmark datasets in Table 2, which verifies that our augmentation method is domain-agnostic. Under the ProjectedGAN framework, we see that with noise properly injected into the high-dimensional feature space, Diffusion ProjectedGAN shows clear improvement in terms of both FID and Recall. We reach state-of-the-art FID results with Diffusion ProjectedGAN on STL-10 and LSUN-Bedroom/Church datasets.

Table 2: Domain-agnostic experiments on ProjectedGAN.

| Domain-agnostic Tasks | CIFAR-10 ($32 \times 32$) | | STL-10 ($64 \times 64$) | | LSUN-Bedroom ($256 \times 256$) | | LSUN-Church ($256 \times 256$) | |
|---|---|---|---|---|---|---|---|---|
| | FID | Recall | FID | Recall | FID | Recall | FID | Recall |
| ProjectedGAN (Sauer et al., 2021) | 3.10 | 0.45 | 7.76 | 0.35 | 2.25 | 0.55 | 3.42 | 0.56 |
| Diffusion ProjectedGAN | **2.54** | 0.45 | **6.91** | 0.35 | **1.43** | **0.58** | **1.85** | **0.65** |

### 4.3 EFFECTIVENESS OF DIFFUSION-GAN FOR LIMITED DATA

We evaluate whether Diffusion-GAN can provide data-efficient GAN training. We first generate five FFHQ ($1024 \times 1024$) dataset splits, consisting of 200, 500, 1k, 2k, and 5k images, respectively, where 200 and 500 images are considered to be extremely limited data cases. We also consider AFHQ-Cat, -Dog, and -Wild ($512 \times 512$), each with as few as around 5k images. Motivated by the success of InsGen (Yang et al., 2021) on small datasets, we build our Diffusion-GAN upon it. We note on limited data, InsGen convincingly outperforms both StyleGAN2+ADA and +DiffAug, and currently holds the state-of-the-art performance for data-efficient GAN training. The results in Table 3 show that our Diffusion-GAN method can help further boost the performance of InsGen in limited data settings.

Table 3: FFHQ ($1024 \times 1024$) FID results with 200, 500, 1k, 2k, and 5k training samples; AFHQ ($512 \times 512$) FID results. To ensure convergence, all models are trained across 10M images for FFHQ and 25M images for AFHQ. We bold the best number in each column.

| Models | FFHQ (200) | FFHQ (500) | FFHQ (1k) | FFHQ (2k) | FFHQ (5k) | Cat | Dog | Wild |
|---|---|---|---|---|---|---|---|---|
| InsGen (Yang et al., 2021) | 102.58 | 54.762 | 34.90 | 18.21 | 9.89 | 2.60* | 5.44* | 1.77* |
| Diffusion InsGen | **63.34** | **50.39** | **30.91** | **16.43** | **8.48** | **2.40** | **4.83** | **1.51** |

## 5 CONCLUSION

We present Diffusion-GAN, a novel GAN framework that uses a variable-length forward diffusion chain with a Gaussian mixture distribution to generate instance noise for GAN training. This approach enables model- and domain-agnostic differentiable augmentation that leverages the advantages of diffusion without requiring a costly reverse diffusion chain. We prove theoretically and demonstrate empirically that Diffusion-GAN can prevent discriminator overfitting and provide non-leaking augmentation. We also demonstrate that Diffusion-GAN can produce high-resolution photo-realistic images with high fidelity and diversity, outperforming its corresponding state-of-the-art GAN baselines on standard benchmark datasets according to both FID and Recall.

ACKNOWLEDGEMENTS

Z. Wang, H. Zheng, and M. Zhou acknowledge the support of NSF-IIS 2212418 and IFML.

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

# Appendix

## A    RELATED WORK

**Stabilizing GAN training.**   A root cause of training difficulties in GANs is often attributed to the JS divergence that GANs intend to minimize. This is because when the data and generator distributions have non-overlapping supports, which are often the case for high-dimensional data supported by low-dimensional manifolds, the gradient of the JS divergence may provide no useful guidance to optimize the generator (Arjovsky and Bottou, 2017; Arjovsky et al., 2017; Mescheder et al., 2018; Roth et al., 2017). For this reason, Arjovsky et al. (2017) propose to instead use the Wasserstein-1 distance, which in theory can provide useful gradient for the generator even if the two distributions have disjoint supports. However, Wasserstein GANs often require the use of a critic function under the 1-Lipschitz constraint, which is difficult to satisfy in practice and hence realized with heuristics such as weight clipping (Arjovsky et al., 2017), gradient penalty (Gulrajani et al., 2017), and spectral normalization (Miyato et al., 2018a).

While the divergence minimization perspective has played an important role in motivating the construction of Wasserstein GANs and gradient penalty-based regularizations, cautions should be made on purely relying on it to understand GAN training, due to not only the discrepancy between the divergence in theory and the actual min-max objective function used in practice, but also the potential confounding between different divergences and different training and regularization strategies (Fedus et al., 2018; Mescheder et al., 2018). E.g., Mescheder et al. (2018) have provided a simple example where in theory the Wasserstein GAN is predicted to succeed while the vanilla GAN is predicted to fail, but in practice the Wasserstein GAN with a finite number of discriminator updates per generator update fails to converge while the vanilla GAN with the non-saturating loss can slowly converge. Fedus et al. (2018) provide a rich set of empirical evidence to discourage viewing GANs purely from the perspective of minimizing a specific divergence at each training step and emphasize the important role played by gradient penalties on stabilizing GAN training.

**Diffusion models.**    Due to the use of a forward diffusion chain, the proposed Diffusion-GAN can be related to diffusion-based (or score-based) deep generative models (Ho et al., 2020b; Sohl-Dickstein et al., 2015; Song and Ermon, 2019) that employ both a forward (inference) and a reverse (generative) diffusion chain. These diffusion-based generative models are stable to train and can generate high-fidelity photo-realistic images (Dhariwal and Nichol, 2021; Ho et al., 2020b; Nichol et al., 2021; Ramesh et al., 2022; Song and Ermon, 2019; Song et al., 2021b). However, they are notoriously slow in generation due to the need to traverse the reverse diffusion chain, which involves going through the same U-Net-based generator network hundreds or even thousands of times (Song et al., 2021a). For this reason, a variety of methods have been proposed to reduce the generation cost of diffusion-based generative models (Kong and Ping, 2021; Luhman and Luhman, 2021; Pandey et al., 2022; San-Roman et al., 2021; Song et al., 2021a; Xiao et al., 2021; Zheng et al., 2022).

A key distinction is that Diffusion-GAN needs a reverse diffusion chain during neither training nor generation. More specifically, its generator maps the noise to a generated sample in a single step. Diffusion-GAN can train and generate as quickly as a vanilla GAN does with the same generator size. For example, it takes around 20 hours to sample 50k images of size $32 \times 32$ from a DDPM (Ho et al., 2020b) on an Nvidia 2080 Ti GPU, but would take less than a minute to do so from Diffusion-GAN.

**Differentiable augmentation.**  As Diffusion-GAN transforms both the data and generated samples before sending them to the discriminator, we can also relate it to differentiable augmentation (Karras et al., 2020a; Zhao et al., 2020) proposed for data-efficient GAN training. Karras et al. (2020a) introduce a stochastic augmentation pipeline with 18 transformations and develop an adaptive mechanism for controlling the augmentation probability. Zhao et al. (2020) propose to use *Color + Translation + Cutout* as differentiable augmentations for both generated and real images.

While providing good empirical results on some datasets, these augmentation methods are developed with domain-specific knowledge and have the risk of leaking augmentation into generation (Karras et al., 2020a). As observed in our experiments, they sometime worsen the results when applied to a new dataset, likely because the risk of augmentation leakage overpowers the benefits of enlarging the training set, which could happen especially if the training set size is already sufficiently large.

By contrast, Diffusion-GAN uses a differentiable forward diffusion process to stochastically transform the data and can be considered as both a domain-agnostic and a model-agnostic augmentation method. In other words, Diffusion-GAN can be applied to non-image data or even latent features, for which appropriate data augmentation is difficult to be defined, and easily plugged into an existing GAN to improve its generation performance. Moreover, we prove in theory and show in experiments that augmentation leakage is not a concern for Diffusion-GAN. Tran et al. (2021) provide a theoretical analysis for deterministic non-leaking transformation with differentiable and invertible mapping functions. Bora et al. (2018) show similar theorems to us for specific stochastic transformations, such as Gaussian Projection, Convolve+Noise, and stochastic Block-Pixels, while our Theorem 2 includes more satisfying possibilities as discussed in Appendix B.

## B   PROOF

*Proof of Theorem 1.* For simplicity, let $\boldsymbol{x} \sim \mathbb{P}_r$, $\boldsymbol{x}_g \sim \mathbb{P}_g$, $\boldsymbol{y} \sim \mathbb{P}_{r',t}$, $\boldsymbol{y}_g \sim \mathbb{P}_{g',t}$, $a_t = \sqrt{\bar{\alpha}_t}$ and $b_t = (1 - \bar{\alpha}_t)\sigma^2$. Then,

$$p_{r',t}(\boldsymbol{y}) = \int_{\mathcal{X}} p_r(\boldsymbol{x})\mathcal{N}(\boldsymbol{y}; a_t\boldsymbol{x}, b_t\boldsymbol{I})d\boldsymbol{x}$$

$$p_{g',t}(\boldsymbol{y}) = \int_{\mathcal{X}} p_g(\boldsymbol{x})\mathcal{N}(\boldsymbol{y}; a_t\boldsymbol{x}, b_t\boldsymbol{I})d\boldsymbol{x}$$

$$\boldsymbol{z} \sim p(\boldsymbol{z}), \boldsymbol{x}_g = g_\theta(\boldsymbol{z}), \boldsymbol{y}_g = a_t\boldsymbol{x}_g + b_t\boldsymbol{\epsilon}, \boldsymbol{\epsilon} \sim p(\boldsymbol{\epsilon})$$

$$\mathcal{D}_f(p_{r',t}(\boldsymbol{y})\|p_{g',t}(\boldsymbol{y})) = \int_{\mathcal{X}} p_{g',t}(\boldsymbol{y})f\left(\frac{p_{r',t}(\boldsymbol{y})}{p_{g',t}(\boldsymbol{y})}\right)d\boldsymbol{y}$$

$$= \mathbb{E}_{\boldsymbol{y}\sim p_{g',t}(\boldsymbol{y})}\left[f\left(\frac{p_{r',t}(\boldsymbol{y})}{p_{g',t}(\boldsymbol{y})}\right)\right]$$

$$= \mathbb{E}_{\boldsymbol{z}\sim p(\boldsymbol{z}),\boldsymbol{\epsilon}\sim p(\boldsymbol{\epsilon})}\left[f\left(\frac{p_{r',t}(a_tg_\theta(\boldsymbol{z}) + b_t\boldsymbol{\epsilon})}{p_{g',t}(a_tg_\theta(\boldsymbol{z}) + b_t\boldsymbol{\epsilon})}\right)\right]$$

Since $\mathcal{N}(\boldsymbol{y}; a_t\boldsymbol{x}, b_t\boldsymbol{I})$ is assumed to be an isotropic Gaussian distribution, for simplicity, in what follows we show the proof in uni-variate Gaussian, which could be easily extended to multi-variate Gaussian by the production rule. We first show that under mild conditions, the $p_{r',t}(y)$ and $p_{g',t}(y)$ are continuous functions over $y$.

$$\lim_{\Delta y\to 0} p_{r',t}(y - \Delta y) = \lim_{\Delta y\to 0}\int_{\mathcal{X}} p_r(x)\mathcal{N}(y - \Delta y; a_tx, b_t)dx$$

$$= \int_{\mathcal{X}} p_r(x)\lim_{\Delta y\to 0}\mathcal{N}(y - \Delta y; a_tx, b_t)dx$$

$$= \int_{\mathcal{X}} p_r(x)\lim_{\Delta y\to 0}\frac{1}{C_1}\exp\left(\frac{((y - \Delta y) - a_tx)^2}{C_2}\right)dx$$

$$= \int_{\mathcal{X}} p_r(x)\mathcal{N}(y; a_tx, b_t)dx$$

$$= p_{r',t}(y),$$

where $C_1$ and $C_2$ are constants. Hence, $p_{r',t}(y)$ is a continuous function defined on $y$. The proof of continuity for $p_{g',t}(y)$ is exactly the same proof. Then, given $g_\theta$ is also a continuous function, it is clear to see that $\mathcal{D}_f(p_{r',t}(\boldsymbol{y})\|p_{g',t}(\boldsymbol{y}))$ is a continuous function over $\theta$.

Next, we show that $\mathcal{D}_f(p_{r',t}(\boldsymbol{y})\|p_{g',t}(\boldsymbol{y}))$ is differentiable. By the chain rule, showing $\mathcal{D}_f(p_{r',t}(\boldsymbol{y})\|p_{g',t}(\boldsymbol{y}))$ to be differentiable is equivalent to show $p_{r',t}(y)$, $p_{r',t}(y)$ and $f$ are differentiable. Usually, $f$ is defined with differentiability (Nowozin et al., 2016).

$$\nabla_\theta p_{r',t}(a_tg_\theta(z) + b_t\epsilon) = \nabla_\theta\int_{\mathcal{X}} p_r(x)\mathcal{N}(a_tg_\theta(z) + b_t\epsilon; a_tx, b_t)dx$$

$$= \int_{\mathcal{X}} p_r(x)\frac{1}{C_1}\nabla_\theta\exp\left(\frac{\|a_tg_\theta(z) + b_t\epsilon - a_tx\|_2^2}{C_2}\right)dx,$$

$$\nabla_\theta p_{g',t}(a_t g_\theta(z) + b_t \epsilon) = \nabla_\theta \int_{\mathcal{X}} p_g(x) \mathcal{N}(a_t g_\theta(z) + b_t \epsilon; a_t x, b_t) dx$$

$$= \nabla_\theta \mathbb{E}_{z' \sim p(z')} \left[ \mathcal{N}(a_t g_\theta(z) + b_t \epsilon; a_t g_\theta(z'), b_t) \right]$$

$$= \mathbb{E}_{z' \sim p(z')} \left[ \frac{1}{C_1} \nabla_\theta \exp \left( \frac{||a_t g_\theta(z) + b_t \epsilon - a_t g_\theta(z')||_2^2}{C_2} \right) \right],$$

where $C_1$ and $C_2$ are constants. Hence, $p_{r',t}(y)$ and $p_{r',t}(y)$ are differentiable, which concludes the proof. $\qquad\square$

*Proof of Theorem 2.* We have $p(\boldsymbol{y}) = \int p(\boldsymbol{x}) q(\boldsymbol{y} \,|\, \boldsymbol{x}) d\boldsymbol{x}$ and $p_g(\boldsymbol{y}) = \int p_g(\boldsymbol{x}) q(\boldsymbol{y} \,|\, \boldsymbol{x}) d\boldsymbol{x}$.
$\Leftarrow$ If $p(\boldsymbol{x}) = p_g(\boldsymbol{x})$, then $p(\boldsymbol{y}) = p_g(\boldsymbol{y})$
$\Rightarrow$ Let $\boldsymbol{y} \sim p(\boldsymbol{y})$ and $\boldsymbol{y}_g \sim p_g(\boldsymbol{y})$. Given the assumption on $q(\boldsymbol{y} \,|\, \boldsymbol{x})$, we have

$$\boldsymbol{y} = f(\boldsymbol{x}) + g(\boldsymbol{\epsilon}), \boldsymbol{x} \sim p(\boldsymbol{x}), \boldsymbol{\epsilon} \sim p(\boldsymbol{\epsilon})$$
$$\boldsymbol{y}_g = f(\boldsymbol{x}_g) + g(\boldsymbol{\epsilon}_g), \boldsymbol{x}_g \sim p_g(\boldsymbol{x}), \boldsymbol{\epsilon}_g \sim p(\boldsymbol{\epsilon}).$$

Since $f$ and $g$ are one-to-one mapping functions, $f(\boldsymbol{x})$ and $g(\boldsymbol{\epsilon})$ are identifiable, which indicates $f(\boldsymbol{x}) \stackrel{D}{=} f(\boldsymbol{x}_g) \Rightarrow \boldsymbol{x} \stackrel{D}{=} \boldsymbol{x}_g$. By the property of moment-generating functions (MGF), given $f(\boldsymbol{x})$ is independent with $g(\boldsymbol{\epsilon})$, we have for $\forall \boldsymbol{s}$

$$M_{\boldsymbol{y}}(\boldsymbol{s}) = M_{f(\boldsymbol{x})}(\boldsymbol{s}) \cdot M_{g(\boldsymbol{\epsilon})}(\boldsymbol{s})$$
$$M_{\boldsymbol{y}_g}(\boldsymbol{s}) = M_{f(\boldsymbol{x}_g)}(\boldsymbol{s}) \cdot M_{g(\boldsymbol{\epsilon}_g)}(\boldsymbol{s}).$$

where $M_{\boldsymbol{y}}(\boldsymbol{s}) = E_{\boldsymbol{y} \sim p(\boldsymbol{y})}[e^{\boldsymbol{s}^T \boldsymbol{y}}]$ denotes the MGF of random variable $\boldsymbol{y}$ and the others follow the same form. By the moment-generating function uniqueness theorem, given $\boldsymbol{y} \stackrel{D}{=} \boldsymbol{y}_g$ and $g(\boldsymbol{\epsilon}) \stackrel{D}{=} g(\boldsymbol{\epsilon}_g)$, we have $M_{\boldsymbol{y}}(\boldsymbol{s}) = M_{\boldsymbol{y}_g}(\boldsymbol{s})$ and $M_{g(\boldsymbol{\epsilon})}(\boldsymbol{s}) = M_{g(\boldsymbol{\epsilon}_g)}(\boldsymbol{s})$ for $\forall \boldsymbol{s}$. Then, we could obtain $M_{f(\boldsymbol{x})} = M_{f(\boldsymbol{x}_g)}$ for $\forall \boldsymbol{s}$. Thus, $M_{f(\boldsymbol{x})} = M_{f(\boldsymbol{x}_g)} \Rightarrow f(\boldsymbol{x}) \stackrel{D}{=} f(\boldsymbol{x}_g) \Rightarrow p(\boldsymbol{x}) = p(\boldsymbol{x}_g)$, which concludes the proof.

**Discussion.** Next, we discuss which $q(\boldsymbol{y} \,|\, \boldsymbol{x})$ fits the assumption we made on it. We follow the discussion of reparameterization of distributions as used in Kingma and Welling (2014). Three basic approaches are:

1. Tractable inverse CDF. In this case, let $\boldsymbol{\epsilon} \sim \mathcal{U}(\boldsymbol{0}, \boldsymbol{I})$, and $\psi(\boldsymbol{\epsilon}, \boldsymbol{y}, \boldsymbol{x})$ be the inverse CDF of $q(\boldsymbol{y} \,|\, \boldsymbol{x})$. From $\psi(\boldsymbol{\epsilon}, \boldsymbol{y}, \boldsymbol{x})$, if $\boldsymbol{y} = f(\boldsymbol{x}) + g(\boldsymbol{\epsilon})$, for example, $y \sim \text{Cauchy}(x, \gamma)$ and $y \sim \text{Logistic}(x, s)$, then Theorem 2 holds.

2. Analogous to the Gaussian example, $\boldsymbol{y} \sim \mathcal{N}(\boldsymbol{x}, \sigma^2 \boldsymbol{I}) \Rightarrow \boldsymbol{y} = \boldsymbol{x} + \sigma \cdot \boldsymbol{\epsilon}, \boldsymbol{\epsilon} \sim \mathcal{N}(\boldsymbol{0}, \boldsymbol{I})$. For any "location-scale" family of distributions we can choose the standard distribution (with location = 0, scale = 1) as the auxiliary variable $\boldsymbol{\epsilon}$, and let $g(.) = \text{location} + \text{scale} \cdot \boldsymbol{\epsilon}$. Examples: Laplace, Elliptical, Student's t, Logistic, Uniform, Triangular, and Gaussian distributions.

3. Implicit distributions. $q(\boldsymbol{y} \,|\, \boldsymbol{x})$ could be modeled by neural networks, which implies $\boldsymbol{y} = f(\boldsymbol{x}) + g(\boldsymbol{\epsilon}), \boldsymbol{\epsilon} \sim p(\boldsymbol{\epsilon})$, where $f$ and $g$ are one-to-one nonlinear transformations.

$\qquad\square$

## C  DERIVATIONS

**Derivation of equality in JSD**

$$JSD(p(\boldsymbol{y},t), p_g(\boldsymbol{y},t))$$
$$= \frac{1}{2}\mathcal{D}_{\text{KL}}\left[p(\boldsymbol{y},t)\Big|\Big|\frac{p(\boldsymbol{y},t)+p_g(\boldsymbol{y},t)}{2}\right] + \frac{1}{2}\mathcal{D}_{\text{KL}}\left[p_g(\boldsymbol{y},t)\Big|\Big|\frac{p(\boldsymbol{y},t)+p_g(\boldsymbol{y},t)}{2}\right]$$
$$= \frac{1}{2}E_{\boldsymbol{y},t\sim p(\boldsymbol{y},t)}\left[\log\frac{2\cdot p(\boldsymbol{y},t)}{p(\boldsymbol{y},t)+p_g(\boldsymbol{y},t)}\right] + \frac{1}{2}E_{\boldsymbol{y},t\sim p_g(\boldsymbol{y},t)}\left[\log\frac{2\cdot p_g(\boldsymbol{y},t)}{p(\boldsymbol{y},t)+p_g(\boldsymbol{y},t)}\right]$$
$$= \frac{1}{2}E_{t\sim p_\pi(t),\boldsymbol{y}\sim p(\boldsymbol{y}\,|\,t)}\left[\log\frac{2\cdot p(\boldsymbol{y}\,|\,t)p_\pi(t)}{p(\boldsymbol{y}\,|\,t)p_\pi(t)+p_g(\boldsymbol{y}\,|\,t)p_\pi(t)}\right]$$
$$+ \frac{1}{2}E_{t\sim p_\pi(t),\boldsymbol{y}\sim p_g(\boldsymbol{y}\,|\,t)}\left[\log\frac{2\cdot p_g(\boldsymbol{y}\,|\,t)p_\pi(t)}{p(\boldsymbol{y}\,|\,t)p_\pi(t)+p_g(\boldsymbol{y}\,|\,t)p_\pi(t)}\right]$$
$$= \mathbb{E}_{t\sim p_\pi(t)}\left[\frac{1}{2}E_{\boldsymbol{y}\sim p(\boldsymbol{y}\,|\,t)}\left[\log\frac{2\cdot p(\boldsymbol{y}\,|\,t)}{p(\boldsymbol{y}\,|\,t)+p_g(\boldsymbol{y}\,|\,t)}\right] + \frac{1}{2}E_{\boldsymbol{y}\sim p_g(\boldsymbol{y}\,|\,t)}\left[\log\frac{2\cdot p_g(\boldsymbol{y}\,|\,t)}{p(\boldsymbol{y}\,|\,t)+p_g(\boldsymbol{y}\,|\,t)}\right]\right]$$
$$= \mathbb{E}_{t\sim p_\pi(t)}[JSD(p(\boldsymbol{y}\,|\,t), p_g(\boldsymbol{y}\,|\,t))].$$

## D  DETAILS OF TOY EXAMPLE

Here, we provide the detailed analysis of the JS divergence toy example.

**Notation.**  Let $\mathcal{X}$ be a compact metric set (such as the space of images $[0,1]^d$) and $\text{Prob}(\mathcal{X})$ denote the space of probability measures defined on $\mathcal{X}$. Let $\mathbb{P}_r$ be the target data distribution and $\mathbb{P}_g$ [1] be the generator distribution. The JSD between the two distributions $\mathbb{P}_r, \mathbb{P}_g \in \text{Prob}(\mathcal{X})$ is defined as:

$$\mathcal{D}_{\text{JS}}(\mathbb{P}_r||\mathbb{P}_g) = \frac{1}{2}\mathcal{D}_{\text{KL}}(\mathbb{P}_r||\mathbb{P}_m) + \frac{1}{2}\mathcal{D}_{\text{KL}}(\mathbb{P}_g||\mathbb{P}_m), \tag{7}$$

where $\mathbb{P}_m$ is the mixture $(\mathbb{P}_r + \mathbb{P}_g)/2$ and $\mathcal{D}_{\text{KL}}$ denotes the Kullback-Leibler divergence, *i.e.*, $\mathcal{D}_{\text{KL}}(\mathbb{P}_r||\mathbb{P}_g) = \int_{\mathcal{X}} p_r(x)\log(\frac{p_r(x)}{p_\theta(x)})dx$. More generally, the $f$-divergence (Nowozin et al., 2016) between $\mathbb{P}_r$ and $\mathbb{P}_g$ is defined as:

$$\mathcal{D}_f(\mathbb{P}_r||\mathbb{P}_g) = \int_{\mathcal{X}} p_g(\boldsymbol{x})f\left(\frac{p_r(\boldsymbol{x})}{p_g(\boldsymbol{x})}\right)d\boldsymbol{x}, \tag{8}$$

where the generator function $f : \mathbb{R}_+ \to \mathbb{R}$ is a convex and lower-semicontinuous function satisfying $f(1) = 0$. We refer to Nowozin et al. (2016) for more details.

We recall the typical example introduced in Arjovsky and Bottou (2017) and follow the notations.

**Example.**  Let $Z \sim U[0,1]$ be the uniform distribution on the unit interval. Let $X \sim \mathbb{P}_r$ be the distribution of $(0, Z) \in \mathbb{R}^2$, which contains a 0 on the x-axis and a random variable Z on the y-axis. Let $X_g \sim \mathbb{P}_g$ be the distribution of $(\theta, Z) \in \mathbb{R}^2$, where $\theta$ is a single real parameter. In this case, the $\mathcal{D}_{\text{JS}}(\mathbb{P}_r||\mathbb{P}_g)$ is not continuous,

$$\mathcal{D}_{\text{JS}}(\mathbb{P}_r||\mathbb{P}_g) = \begin{cases} 0 & \text{if } \theta = 0, \\ \log 2 & \text{if } \theta \neq 0. \end{cases}$$

---

[1]For notation simplicity, $g$ and $G$ both denote the generator network in GANs in this paper.

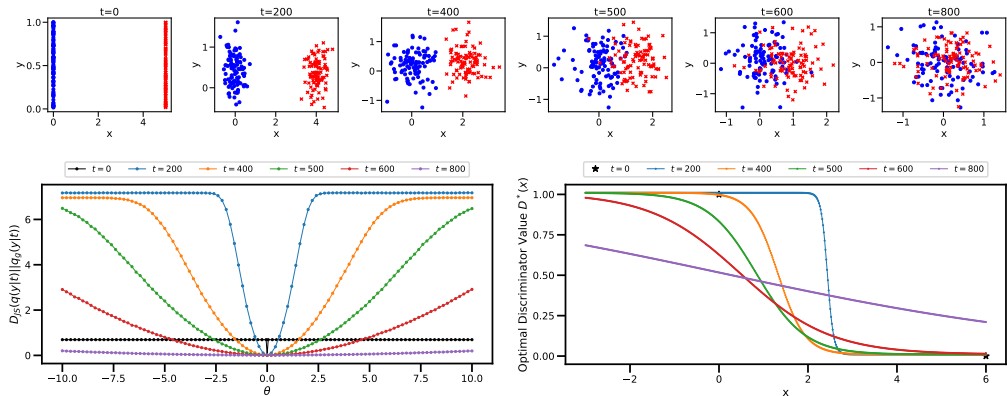

Figure 6: We show the data distribution and $\mathcal{D}_{JS}(\mathbb{P}_r||\mathbb{P}_g)$.

which can not provide a usable gradient for training. The derivation is as follows:

$$\mathcal{D}_{JS}(\mathbb{P}_r||\mathbb{P}_g) = \frac{1}{2}\mathbb{E}_{x\sim p_r(x)}\left[\log\frac{2\cdot p_r(x)}{p_r(x)+p_g(x)}\right] + \frac{1}{2}\mathbb{E}_{y\sim p_g(y)}\left[\log\frac{2\cdot p_g(y)}{p_r(y)+p_g(y)}\right]$$

$$= \frac{1}{2}\mathbb{E}_{x_1=0,x_2\sim U[0,1]}\left[\log\frac{2\cdot\mathbf{1}[x_1=0]\cdot U(x_2)}{\mathbf{1}[x_1=0]\cdot U(x_2)+\mathbf{1}[x_1=\theta]\cdot U(x_2)}\right]$$

$$+ \frac{1}{2}\mathbb{E}_{y_1=\theta,y_2\sim U[0,1]}\left[\log\frac{2\cdot\mathbf{1}[y_1=\theta]\cdot U(y_2)}{\mathbf{1}[y_1=0]\cdot U(y_2)+\mathbf{1}[y_1=\theta]\cdot U(y_2)}\right]$$

$$= \frac{1}{2}\left[\log\frac{2\cdot\mathbf{1}[x_1=0]}{\mathbf{1}[x_1=0]+\mathbf{1}[x_1=\theta]}\Big|x_1=0\right] + \frac{1}{2}\left[\log\frac{2\cdot\mathbf{1}[y_1=\theta]}{\mathbf{1}[y_1=0]+\mathbf{1}[y_1=\theta]}\Big|y_1=\theta\right]$$

$$= \begin{cases} 0 & \text{if } \theta=0, \\ \log 2 & \text{if } \theta\neq 0. \end{cases}$$

Although this simple example features distributions with disjoint supports, the same conclusion holds when the supports have a non empty intersection contained in a set of measure zero (Arjovsky and Bottou, 2017). This happens to be the case when two low dimensional manifolds intersect in general position (Arjovsky and Bottou, 2017). To avoid the potential issue caused by having non-overlapping distribution supports, a common remedy is to use Wasserstein-1 distance which in theory can still provide usable gradient (Arjovsky and Bottou, 2017; Arjovsky et al., 2017). In this case, the Wasserstein-1 distance is $|\theta|$.

**Diffusion-based noise injection**   In general, with our diffusion noise injected, we could have,

$$p_{r',t} = \int_{\mathcal{X}} p_r(\boldsymbol{x})\mathcal{N}(\boldsymbol{y};\sqrt{\bar{\alpha}_t}\boldsymbol{x},(1-\bar{\alpha}_t)\sigma^2\boldsymbol{I})d\boldsymbol{x}$$

$$p_{g',t} = \int_{\mathcal{X}} p_g(\boldsymbol{x})\mathcal{N}(\boldsymbol{y};\sqrt{\bar{\alpha}_t}\boldsymbol{x},(1-\bar{\alpha}_t)\sigma^2\boldsymbol{I})d\boldsymbol{x}$$

$$\mathcal{D}_{JS}(p_{r',t}||p_{g',t}) = \frac{1}{2}\mathbb{E}_{p_{r',t}}\left[\log\frac{2p_{r',t}}{p_{r',t}+p_{g',t}}\right] + \frac{1}{2}\mathbb{E}_{p_{g',t}}\left[\log\frac{2p_{g',t}}{p_{r',t}+p_{g',t}}\right]$$

For the previous example, we have $Y_t'$ and $Y_{g,t}'$ such that,

$$Y_t'=(y_1,y_2)\sim p_{r',t}=\mathcal{N}(y_1\,|\,0,b_t)f(y_2), Y_{g,t}'=(y_{g,1},y_{g,2})\sim p_{g',t}=\mathcal{N}(y_{g,1}\,|\,a_t\theta,b_t)f(y_{g,2}),$$

where $f(\cdot)=\int_0^1\mathcal{N}(\cdot\,|\,a_tZ,b_t)U(Z)dZ$, $a_t$ and $b_t$ are abbreviations for $\sqrt{\bar{\alpha}_t}$ and $(1-\bar{\alpha}_t)\sigma^2$. The supports of $Y_t'$ and $Y_{g,t}'$ are both the whole metric space $\mathbb{R}^2$ and they overlap with each other depending on $t$, as shown in Figure 2. As $t$ increases, the high density region of $Y_t'$ and $Y_{g,t}'$ get closer

since the weight $a_t$ is decreasing towards 0. Then, we derive the JS divergence,

$$
\begin{aligned}
& \mathcal{D}_{JS}(p_{r',t}||p_{g',t}) \\
&= \frac{1}{2}\mathbb{E}_{y_1\sim\mathcal{N}(y_1\,|\,0,b_t),y_2\sim f(y_2)}\left[\log\frac{2\cdot\mathcal{N}(y_1\,|\,0,b_t)f(y_2)}{\mathcal{N}(y_1\,|\,0,b_t)f(y_2)+\mathcal{N}(y_1\,|\,a_t\theta,b_t)f(y_2)}\right] \\
&\quad+\frac{1}{2}\mathbb{E}_{y_{g,1}\sim\mathcal{N}(y_{g,1}\,|\,0,b_t),y_{g,2}\sim f(y_{g,2})}\left[\log\frac{2\cdot\mathcal{N}(y_{g,1}\,|\,a_t\theta,b_t)f(y_{g,2})}{\mathcal{N}(y_{g,1}\,|\,0,b_t)f(y_{g,2})+\mathcal{N}(y_{g,1}\,|\,a_t\theta,b_t)f(y_{g,2})}\right] \\
&= \frac{1}{2}\mathbb{E}_{y_1\sim\mathcal{N}(0,b_t)}\left[\log\frac{2\cdot\mathcal{N}(y_1\,|\,0,b_t)}{\mathcal{N}(y_1\,|\,0,b_t)+\mathcal{N}(y_1\,|\,a_t\theta,b_t)}\right] \\
&\quad+\frac{1}{2}\mathbb{E}_{y_{g,1}\sim\mathcal{N}(a_t\theta,b_t)}\left[\log\frac{2\cdot\mathcal{N}(y_{g,1}\,|\,a_t\theta,b_t)}{\mathcal{N}(y_{g,1}\,|\,0,b_t)+\mathcal{N}(y_{g,1}\,|\,a_t\theta,b_t)}\right]
\end{aligned}
$$

which is clearly continuous and differentiable.

We show this $\mathcal{D}_{JS}(p_{r',t}||p_{g',t})$ with respect to increasing $t$ values and a $\theta$ grid in the second row of Figure 2. As shown in the left panel, the black line with $t=0$ shows the origianl JSD, which is not even continuous, while as the diffusion level $t$ increments, the lines become smoother and flatter. It is clear to see that these smooth curves provide good learning signals for $\theta$. Recall that the Wasserstein-1 distance is $|\theta|$ in this case. Meanwhile, we could observe with an intense diffusion, $e.g.$, $t=800$, the curve becomes flatter, which indicates smaller gradients and a much slower learning process. This motivates us that an adaptive diffusion could provide different level of gradient smoothness and is possibly better for training. The right panel shows the optimal discriminator outputs over the space $\mathcal{X}$. With diffusion, the optimal discriminator is well defined over the space and the gradient is smooth, while without diffusion the optimal discriminator is only valid on two star points. Interestingly, we find that smaller $t$ drives the optimal discriminator to become more assertive while larger $t$ makes discriminator become more neutral. The diffusion here works like a scale to balance the power of the discriminator.

## E    DATASET DESCRIPTIONS

The CIFAR-10 dataset consists of 50k $32\times32$ training images in 10 categories. The STL-10 dataset originated from ImageNet (Deng et al., 2009) consists of 100k unlabeled images in 10 categories, and we resize them to $64\times64$ resolution. For LSUN datasets, we sample 200k images from LSUN-Bedroom, use the whole 125k images from LSUN-Church, and resize them to $256\times256$ resolution for training. The AFHQ datasets includes around 5k $512\times512$ images per category for dogs, cats, and wild life; we train a separate network for each of them. The FFHQ contains 70k images crawled from Flickr at $1024\times1024$ resolution and we use all of them for training.

## F    ALGORITHM

We provide the Diffusion-GAN algorithm in Algorithm 1.

## G    HYPERPARAMETERS

Diffusion-GAN is built on GAN backbones, so we keep the learning hyperparameters of the original GAN backbones untouched. Diffusion-GAN introduces four new hyperparameters: noise standard deviation $\sigma$, $T_{max}$, $T$ increasing threshold $d_{target}$, and $t$ sampling distribution $p_\pi$.

The $\sigma$ is fixed as 0.05 for images (pixel values rescaled to [-1 ,1]) in all our experiments and it shows good performance. $T_{max}$ could be fixed as 500 or 1000, which depends on the diversity of the dataset. We recommend a large $T_{max}$ for diverse datasets. $d_{target}$ is usually fixed as 0.6, which does not influence much about the performance. $p_\pi$ has two choices, 'uniform' and 'priority'. Generally, $(\sigma=0.05, T_{max}=500, d_{target}=0.6, p_\pi=$ 'uniform') is a good starting point for a new dataset.

In our experiment, we find StyleGAN2-based models are not sensitive to the values of $d_{target}$, so we set $d_{target}=0.6$ for them across all dataset, only except that we set $d_{target}=0.8$ for FFHQ

---

**Algorithm 1** Diffusion-GAN

**while** i $\leq$ number of training iterations **do**

*Step I: Update discriminator*
- Sample minibatch of $m$ noise samples $\{z_1, z_2, \ldots, z_m\} \sim p_z(z)$.
- Obtain generated samples $\{x_{g,1}, x_{g,2}, \ldots, x_{g,m}\}$ by $x_g = G(z)$.
- Sample minibatch of $m$ data examples $\{x_1, x_2, \ldots, x_m\} \sim p(x)$.
- Sample $\{t_1, t_2, \ldots, t_m\}$ from $t_{epl}$ list uniformly with replacement.
- For $j \in \{1, 2, \ldots, m\}$, sample $y_j \sim q(y_j | x_j, t_j)$ and $y_{g,j} \sim q(y_{g,j} | x_{g,j}, t_j)$
- Update discriminator by maximizing Equation (3).

*Step II: Update generator*
- Sample minibatch of $m$ noise samples $\{z_1, z_2, \ldots, z_m\} \sim p_z(z)$
- Obtain generated samples $\{x_{g,1}, x_{g,2}, \ldots, x_{g,m}\}$ by $x_g = G(z)$.
- Sample $\{t_1, t_2, \ldots, t_m\}$ from $t_{epl}$ list with replacement.
- For $j \in \{1, 2, \ldots, m\}$, sample $y_{g,j} \sim q(y_{g,j} | x_{g,j}, t_j)$
- Update generator by minimizing Equation (3).

*Step III: Update diffusion*

**if** i mod 4 == 0 **then**

Update T by Equation (5)

Sample $t_{epl} = [0, \ldots, 0, t_1, \ldots, t_{32}]$, where $t_k \sim p_\pi$ for $k \in \{1, \ldots, 32\}$. $p_\pi$ is in Equation (6). $\{t_{epl}$ has 64 dimensions.$\}$

**end if**

**end while**

---

| Datasets | $r_d$ |
|---|---|
| CIFAR-10 ($32 \times 32$, 50k images) | 0.45 |
| STL-10 ($64 \times 64$, 100k images) | 0.6 |
| LSUN-Church ($256 \times 256$, 120k images) | 0.2 |
| LSUN-Bedroom ($256 \times 256$, 200k images) | 0.2 |

Table 4: $d_{target}$ for Diffusion ProjectedGAN

($d_{target} = 0.8$ for FFHQ is slightly better than 0.6 in FID). We report $d_{target}$ of Diffusion ProjectedGAN for our experiments in Table 4. We also evaluated two $t$ sampling distribution $p_\pi$, ['priority', 'uniform'], defined in Equation (6). In most cases, 'priority' works slightly better, while in some cases, such as FFHQ, 'uniform' is better. Overall, we didn't modify anything in the model architectures and training hyperparameters, such as learning rate and batch size. The forward diffusion configuration and model training configurations are as follows.

**Diffusion config.** For our diffusion-based noise injection, we set up a linearly increasing schedule for $\beta_t$, where $t \in \{1, 2, \ldots, T\}$. For pixel level injection in StyleGAN2, we follow Ho et al. (2020b) and set $\beta_0 = 0.0001$ and $\beta_T = 0.02$. We adaptively modify $T$ ranging from $T_{\min} = 5$ to $T_{\max} = 1000$. The image pixels are usually rescaled to $[-1, 1]$ so we set the Guassian noise standard deviation $\sigma = 0.05$. For feature level injection in Diffusion ProjectedGAN, we set $\beta_0 = 0.0001$, $\beta_T = 0.01$, $T_{\min} = 5$, $T_{\max} = 500$, and $\sigma = 0.5$. We list all these values in Table 5

**Model config.** For StyleGAN2-based models, we borrow the config settings provided by Karras et al. (2020a), which include ['auto', 'stylegan2', 'cifar', 'paper256', 'paper512', 'stylegan2']. We create the 'stl' config based on 'cifar' with a small modification that we change the gamma term to be 0.01. For ProjectedGAN models, we use the recommended default config (Sauer et al., 2021), which is based on FastGAN (Liu et al., 2020). We report the config settings used for our experiments in Table 6.

| Diffusion config for pixel, priority | $\beta_0 = 0.0001, \beta_T = 0.02, T_{\min} = 5, T_{\max} = 1000, \sigma = 0.05$ |
|---|---|
| Diffusion config for pixel, uniform | $\beta_0 = 0.0001, \beta_T = 0.02, T_{\min} = 5, T_{\max} = 500, \sigma = 0.05$ |
| Diffusion config for feature | $\beta_0 = 0.0001, \beta_T = 0.01, T_{\min} = 5, T_{\max} = 500, \sigma = 0.5$ |

Table 5: Diffusion config.

| Dataset | Models | Config | Specification |
|---|---|---|---|
| CIFAR-10 $(32 \times 32)$ | StyleGAN2
Diffusion StyleGAN2
ProjectedGAN
Diffusion ProjectedGAN | cifar
cifar
default
default | -
diffusion-pixel, $d_{target} = 0.6$, 'priority'
diffusion-feature
diffusion-feature |
| STL-10 $(64 \times 64)$ | StyleGAN2
Diffusion StyleGAN2
ProjectedGAN
Diffusion ProjectedGAN | stl
stl
default
default | -
diffusion-pixel, $d_{target} = 0.6$, 'priority'
diffusion-feature
diffusion-feature |
| LSUN-Bedroom $(256 \times 256)$ | StyleGAN2
Diffusion StyleGAN2
ProjectedGAN
Diffusion ProjectedGAN | paper256
paper256
default
default | -
diffusion-pixel, $d_{target} = 0.6$, 'priority'
diffusion-feature
diffusion-feature |
| LSUN-Church $(256 \times 256)$ | StyleGAN2
Diffusion StyleGAN2
ProjectedGAN
Diffusion ProjectedGAN | paper256
paper256
default
default | -
diffusion-pixel, $d_{target} = 0.6$, 'priority'
diffusion-feature
diffusion-feature |
| AFHQ-Cat/Dog/Wild $(512 \times 512)$ | StyleGAN2
Diffusion StyleGAN2
InsGen
Diffusion InsGen | paper512
paper512
default
paper512 | -
diffusion-pixel, $d_{target} = 0.6$, 'priority'
-
diffusion-pixel, $d_{target} = 0.6$, 'uniform' |
| FFHQ $(1024 \times 1024)$ | StyleGAN2
Diffusion StyleGAN2
InsGen
Diffusion InsGen | stylegan2
stylegan2
default
stylegan2 | -
diffusion-pixel, $d_{target} = 0.8$, 'uniform'
-
diffusion-pixel, $d_{target} = 0.6$, 'uniform' |

Table 6: The config setting of StyleGAN2-based models and ProjectedGAN-based models. For StyleGAN2-based models, we borrow the config settings provided by Karras et al. (2020a), which includes ['auto', 'stylegan2', 'cifar', 'paper256', 'paper512', 'paper1024']. We create the 'stl' config based on 'cifar' with small modifications that we change the gamma term to be 0.01. For ProjectedGAN models, we use the recommended default config (Sauer et al., 2021), which is based on FastGAN.

## H  Implementation details

We implement an additional diffusion sampling pipeline, where the diffusion configurations are set in Appendix G. The $T$ in the forward diffusion process is adaptively adjusted and clipped to $[T_{\min}, T_{\max}]$. As illustrated in Algorithm 1, at each update step, we sample $t$ from $t_{epl}$ for each data point $x$, and then use the analytic Gaussian distribution at diffusion step $t$ to sample $y$. Next, we use $y$ and $t$ instead of $x$ for optimization.

**Diffusion StyleGAN2.**  We inherit all the network architectures from StyleGAN2 implemented by Karras et al. (2020a). We modify the original mapping network, which is there for label conditioning and unused for unconditional image generation tasks, inside the discriminator to inject $t$. Specifically, we change the original input of mapping network, the class label $c$, to our discrete value timestep $t$. Then, we train the generator and discriminator with diffused samples $y$ and $t$.

**Diffuson ProjectedGAN.**  To simplify the implementation and minimize the modifications to ProjectedGAN, we construct the discriminator as $D_\phi(y)$, where $t$ is ignored. Our method is plugged in as a data augmentation method. The only change in the optimization stage is that the discriminator is fed with diffused images $y$ instead of original images $x$.

**Diffuson InsGen.**  To simplify the implementation and minimize the modifications to InsGen, we keep their contrastive learning part untouched. We modify the original discriminator network

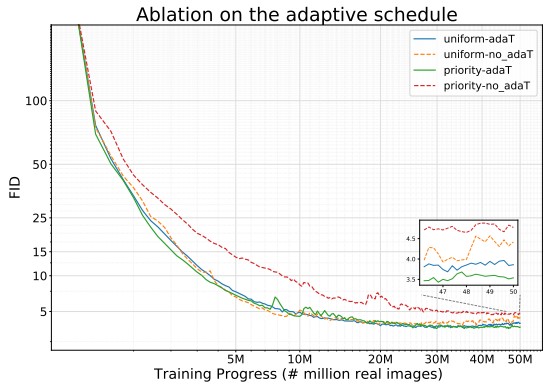

Figure 7: Ablation study on the $T$ adaptiveness.

to inject $t$ similarly to Diffusion StyleGAN2. Then, we train the generator and discriminator with diffused samples $y$ and $t$.

## I    ABLATION ON THE MIXING PROCEDURE AND $T$ ADAPTIVENESS

Note the mixing procedure described in Equation (6), referred to as "priority mixing" in what follows, is designed based on our intuition. Here we conduct an ablation study on the mixing procedure by comparing the priority mixing with uniform mixing on three representative datasets. We report in Table 7 the FID results, which suggest that uniform mixing could work better than priority mixing in some dataset, and hence Diffusion-GAN may be further improved by optimizing its mixing procedure according to the training data. While optimizing the mixing procedure is beyond the focus of this paper, it is worth further investigation in future studies.

Table 7: Ablation study on the mixing procedure. "Priority Mixing" refers to the mixing procedure in Equation (6) and "Uniform Mixing" refers to sample $t$ uniformly at random.

|  | CIFAR-10 | STL-10 | FFHQ |
|---|---|---|---|
| Priority Mixing | **3.19** | **11.43** | 3.22 |
| Uniform Mixing | 3.44 | 11.75 | **2.83** |

We further conduct ablation study on whether the $T$ needs to be adaptively adjusted. As shown in Figure 7, we observe with adaptive diffusion strategy, the training curves of FIDs converge faster and reach lower final FIDs.

## J    MORE GAN VARIANTS

To further validate our noise injection via diffusion-based mixtures, we add our diffusion-based training into two more representative GAN variants: DCGAN (Radford et al., 2015) and SNGAN (Miyato et al., 2018b), which have quite different GAN architectures compared to StyleGAN2. We provide the FIDs for CIFAR-10 in Table 8. We observe that both Diffusion-DCGAN and Diffusion-SNGAN clearly outperform their corresponding baseline GANs.

Table 8: FIDs on CIFAR-10 for DCGAN, Diffusion-DCGAN, SNGAN, and Diffusion-SNGAN.

|  | DCGAN (Radford et al., 2015) | Diffusion-DCGAN | SNGAN (Miyato et al., 2018b) | Diffusion-SNGAN |
|---|---|---|---|---|
| CIFAR-10 | 28.65 | **24.67** | 20.76 | **17.23** |

| Method | IS ↑ | FID ↓ | Recall ↑ | NFE ↓ |
|---|---|---|---|---|
| DDPM (Ho et al., 2020a) | 9.46 | 3.21 | 0.57 | 1000 |
| DDIM (Song et al., 2020) | 8.78 | 4.67 | 0.53 | 50 |
| Denoising Diffusion GAN (Xiao et al., 2021) | 9.63 | 3.75 | 0.57 | 4 |
| StyleGAN2 (Karras et al., 2020a) | 9.18 | 8.32 | 0.41 | 1 |
| StyleGAN2 + DiffAug (Zhao et al., 2020) | 9.40 | 5.79 | 0.42 | 1 |
| StyleGAN2 + ADA (Karras et al., 2020a) | 9.83 | **2.92** | 0.49 | 1 |
| Diffusion StyleGAN2 | **9.94** | 3.19 | **0.58** | 1 |

Table 9: Inception Score for CIFAR-10. For sampling time, we use the number of function evaluations (NFE).

## K    INCEPTION SCORE FOR CIFAR-10

We report the Inception Score (IS) (Salimans et al., 2016) of Diffusion StyleGAN2 for CIFAR-10 dataset in Table 9 and also include other state-of-the-art GANs and diffusion models as baselines. Note CIFAR-10 is a well-known dataset and tested by almost all baselines, so we pick CIFAR-10 here and we reference the reported IS values from their original papers for a fair comparison.

## L    MORE GENERATED IMAGES

We provide more randomly generated images for LSUN-Bedroom, LSUN-Church, AFHQ, and FFHQ datasets in Figure 8, Figure 9, and Figure 10.

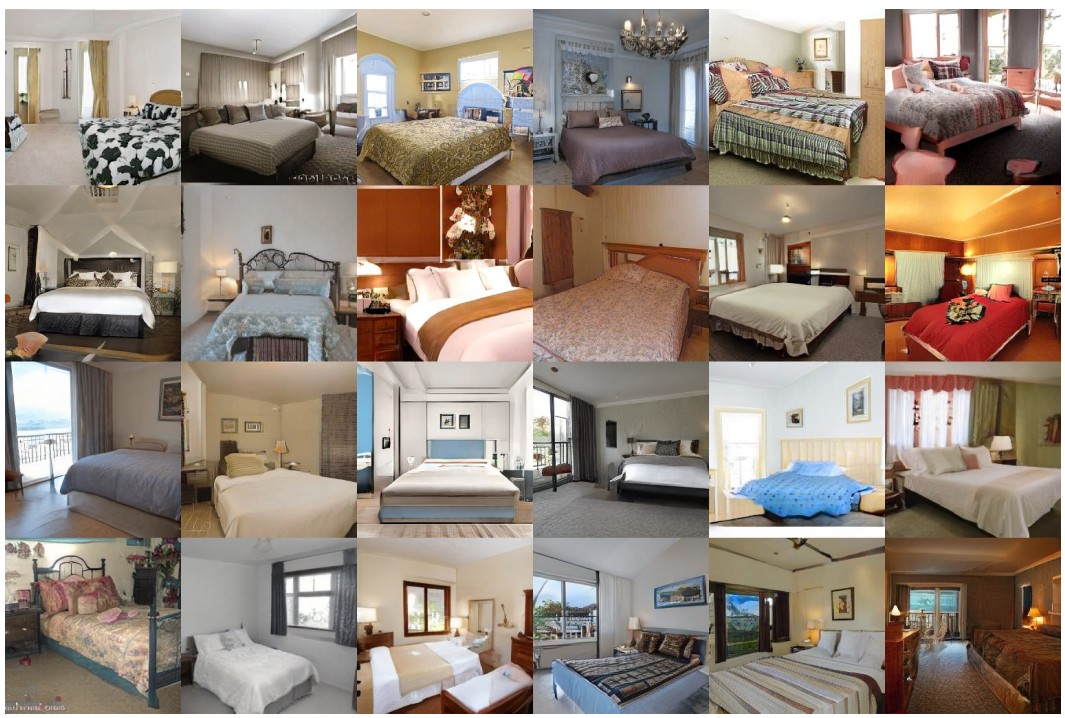

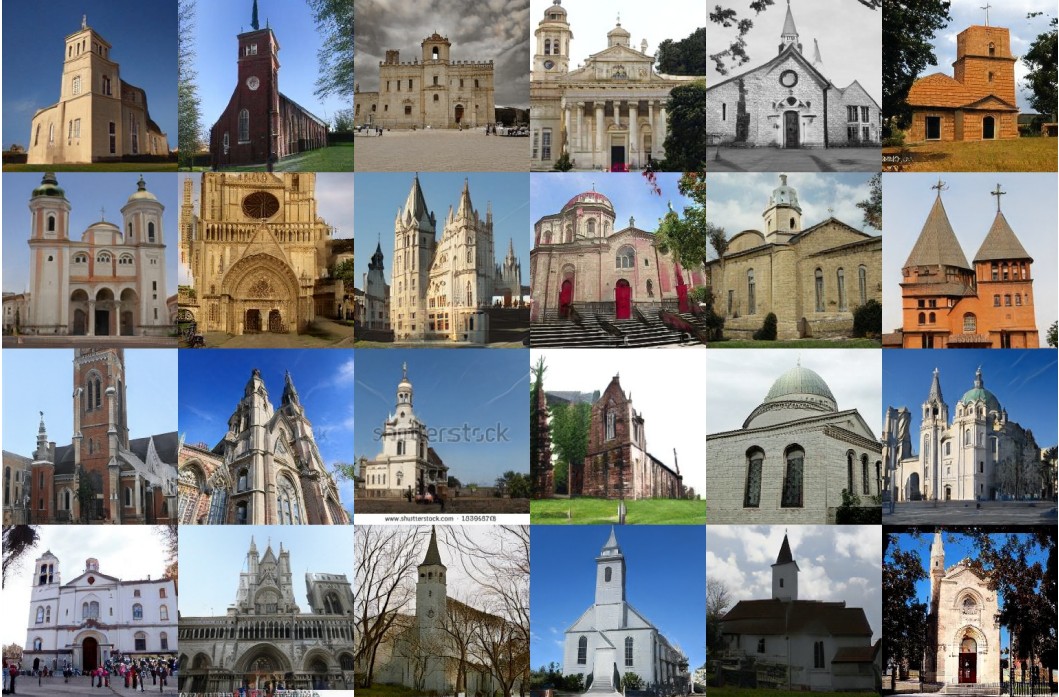

Figure 8: More generated images for LSUN-Bedroom (FID 1.43, Recall 0.58) and LSUN-Church (FID 1.85, Recall 0.65) from Diffusion ProjectedGAN.

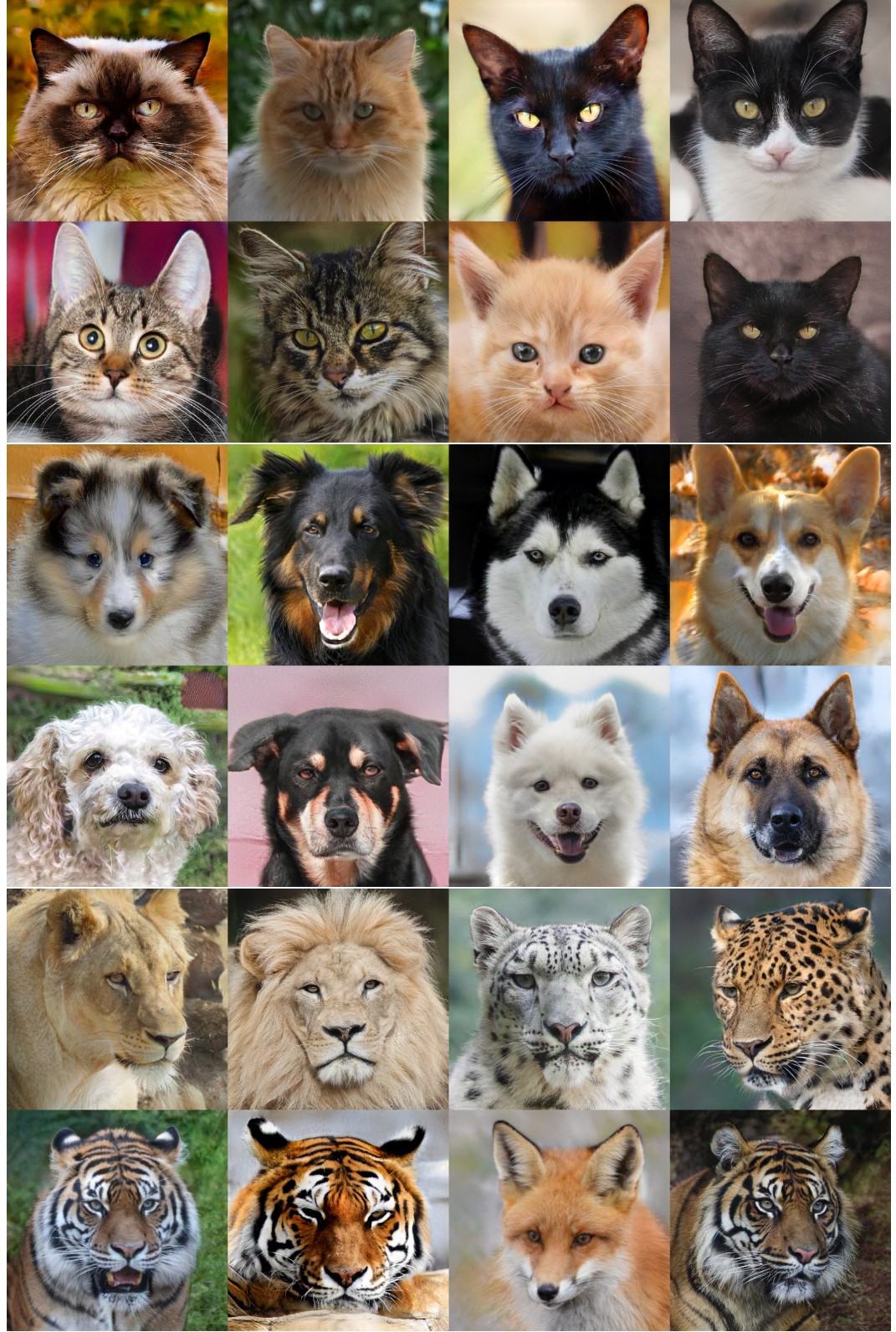

Figure 9: More generated images for AFHQ-Cat (FID 2.40), AFHQ-Dog (FID 4.83) and AFHQ-Wild (FID 1.51) from Diffusion InsGen.

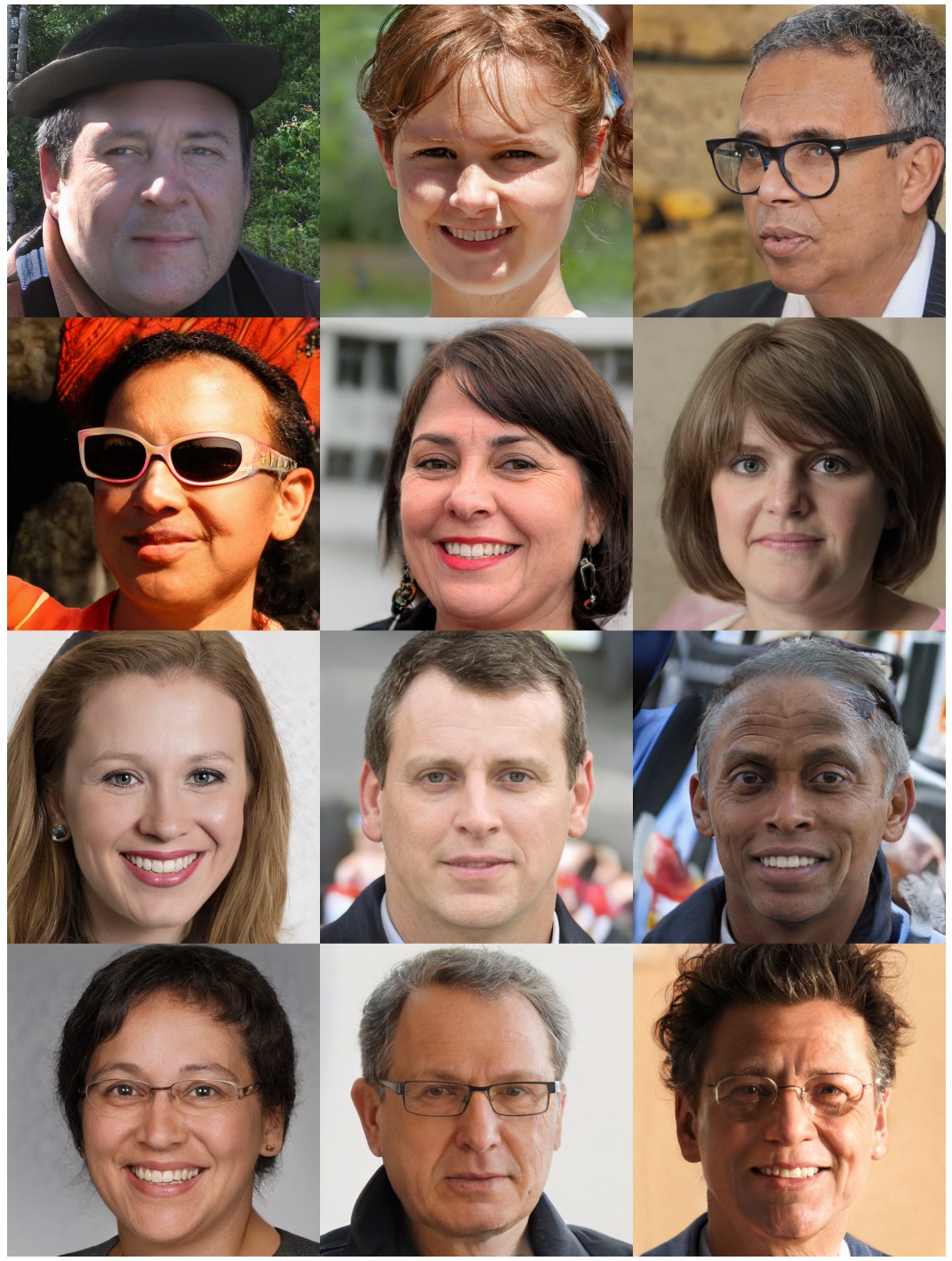

Figure 10: More generated images for FFHQ from Diffusion StyleGAN2 (FID 3.71, Recall 0.43).

