# OpenReview forum: "Diffusion-GAN: Training GANs with Diffusion"
_ICLR.cc/2023/Conference — ICLR 2023 poster_

### Official Review · Reviewer_xwap · 2022-10-22

**Confidence:** 4
**Correctness:** 4
**Technical Novelty And Significance:** 3
**Empirical Novelty And Significance:** 3
**Recommendation:** 6

**Clarity, Quality, Novelty And Reproducibility:**

Clarity: High.

Quality: High. Method and experiments are well written with high quality.

Novelty: Fair.

Reproducibility: High.

**Strength And Weaknesses:**

Strength
1. The paper is clearly written with theoretical analysis to support proposed method.
2. Experiments are well designed and demonstrated clear improvments over baseline methods, ranging from low-reoslution to high-resolution and from low-diversity to high-diversity

Weakness
The method improvment is somewhat marginal: introducing noise to discriminator training has been explored before, introducing different levels of noises as defined in diffusion process seems like a minor improvement.

**Summary Of The Paper:**

This paper proposes to stabalize GAN training by training Discriminator with adapitvely noised images. The method shows improved performance over GAN baselines across low-resolution and high-resolution image generation benchmarks as well as low and high data regime.

**Summary Of The Review:**

The paper is well structured with theoretical analysis and convincing experiments. Introducing adaptive noise as formulated in diffusion process to GAN Discriminator training is somewhat a natural extension of previous works but could be a go-to solution to stabilize GAN  training in the future.

---

> ### Author Response · Authors · 2022-11-11
> **Response**
>
> We thank reviewer xwap for the positive feedback and comments. We further provide clarifications on our motivations and contributions below.
>
> Although stabilizing GAN training through injecting instance noise has been explored before, to the best of our knowledge, there is no existing work that is able to empirically demonstrate the success of using instance noise in GAN training on high-dimensional image data. Roth et al. (2017) show that adding instance noise to the high-dimensional discriminator input does not work well. Our work is the first work that both theoretically justifies it and solves the problem empirically. We think that our noise injection through a mixture distribution defined over the forward diffusion chain is new and novel, which stabilizes the GAN training.
>
> On the other perspective, our work could be seen as one possible bridge between diffusion and GAN methods. We train the generator $G$ on the whole forward diffusion chain and the discriminator $D(\mathbf{x}, t)$ **depends on the diffusion timestep $t$** to distinguish a diffused true image and a diffused fake image. Note this is different from both the original GANs and diffusion models, and we believe this is the **first work** to integrate them to achieve state-of-the-art generation results without increasing the cost of generation. By minimizing $E_{t}[\mathcal{D}_{JS}(p(y | t) || p_g(y || t))]$, which is somewhat related to the diffusion model objective, empirically Diffusion-GAN achieves high fidelity and diverse samples while keeping the sampling time as fast as vanilla GANs.
>
> We hope that our response will clarify the reviewer's concerns and convince reviewer xwap of the significance and novelty of our work.
>
> Kevin Roth, Aurelien Lucchi, Sebastian Nowozin, and Thomas Hofmann. Stabilizing training of generative adversarial networks through regularization. Advances in neural information processing systems, 30, 2017.

---

### Official Review · Reviewer_dVvW · 2022-10-22

**Confidence:** 3
**Correctness:** 3
**Technical Novelty And Significance:** 3
**Empirical Novelty And Significance:** 3
**Recommendation:** 6

**Clarity, Quality, Novelty And Reproducibility:**

- The paper is in general easy to follow.
- As far as I am aware, the approach is original; however, given the ubiquity of diffusion-based approaches in recent years, I might be missing some recent similar efforts in this area.
- Source code is provided.

**Strength And Weaknesses:**

Strengths:
- The proposed approach makes intuitive sense and it is explained with sufficient clarity.
- Generation performance is superior to relevant baselines.
- Computational costs are on a par with relevant baselines (StyleGAN-ADA).
- Some theoretical support is provided to ensure the injection of extra noise does not hamper the basic GAN learning properties (though it is difficult to assess the ultimate significance of the provided theorems)

Weaknesses:
- Claims of stable training were made, but I did not easily find where this was justified. Perhaps the authors could point me to where it is shown that the training is especially stable, in comparison to baselines?
- It was not clear which hyper-parameters need to be addressed while switching the datasets. Again, a simple summary clarification from the authors might help here.

**Summary Of The Paper:**

The paper proposes an improved GAN training method that involves a diffusion chain that generates Gaussian mixture distributed instance noise. The diffusion level is dependent on timestep, and the discrimination task is made increasingly difficult for the discriminator over the course of training. The approach improves generation quality in terms of FID measurements and recall capacity, while allegedly retaining stability.

**Summary Of The Review:**

The paper contributes a non-trivial new approach to GAN training. It could prove a useful addition to the GAN models and provides one possible bridge between diffusion and GAN methods.

---

> ### Author Response · Authors · 2022-11-11
> **Response**
>
> We thank reviewer dVvW for the positive feedback and instructive comments. We provide clarifications below to answer your two questions.
>
> > Claims of stable training were made, but I did not easily find where this was justified. Perhaps the authors could point me to where it is shown that the training is especially stable, in comparison to baselines?
>
> We acknowledge that it is hard to quantify the training stability. In our paper, based on our theory, we claim that by using our method, the discriminator could provide consistently useful learning signals to guide the learning of the generator so as to stabilize the training. Hence, we show the discriminator outputs in Figs. 4 and 5. We observe that the real and fake output distributions of the discriminator from DiffusionGAN overlaps with each other during training, while those from vallina GAN (Fig. 5) could easily deviate from each other. The overlapping in the real and fake distributions shows that the discriminator is not overfitted and keeps improving the generator.
>
>
> > It was not clear which hyper-parameters need to be addressed while switching the datasets. Again, a simple summary clarification from the authors might help here.
>
> Thanks for the great suggestion. We made this more clear in our revision in Appendix G. DiffusionGAN is built on GAN backbones, so we keep the learning hyperparameters of the original GAN backbones untouched. Applying diffusion-based noise injection, we introduce four hyperparameters: noise standard deviation $\sigma$, $T_{max}$, $T$ increasing threshold $d_{target}$, and $t$ sampling distribution $p_{\pi}$.
>
> The $\sigma$ is fixed as 0.05 for images (pixel values rescaled to [-1 ,1]) in all our experiments and it shows good performance. $T_{max}$ could be fixed as 500 or 1000, which depends on the diversity of the dataset. We recommend a large $T_{max}$ for diverse datasets. $d_{target}$ is usually fixed as 0.6, which does not influence much about the performance. $p_{\pi}$ has two choices, 'uniform' and 'priority'. Generally, $(\sigma=0.05, T_{max}=500, d_{target}=0.6, p_{\pi}=\text{`uniform'})$ is a good starting point for a new dataset.

---

> > ### Comment · Reviewer_dVvW · 2022-11-27
> > **Concluding thoughts**
> >
> > I thank the authors for the clarifications and extra information on both accounts. I agree that the distribution overlap argument is a necessary condition for training stability, though I am not completely convinced that it is strong enough to warrant presenting stability of training as an explicit virtue of this method.
> >
> > However, this is a relatively small matter, and I consider my concerns addressed. I continue to favor acceptance.

---

### Official Review · Reviewer_YnnY · 2022-10-25

**Confidence:** 4
**Correctness:** 3
**Technical Novelty And Significance:** 4
**Empirical Novelty And Significance:** 3
**Recommendation:** 8

**Clarity, Quality, Novelty And Reproducibility:**

The paper is novel by combining two model training process and is of good clarity. It should also be easy to reproduce.

**Strength And Weaknesses:**

The main strength:
1. Proposed a new architecture which combines two state-of-the-art generative models and theorectially showed the ability of acheiving a better GAN training and also sampling;
2. The experiments are well-designed and the results are significant and interesting;
3. The paper is well-written and easy to follow

Concerns/questions:
1. While the FID and Recall are reported, is it possible to also include other metrics(IS, LPIPS, ..);


**Summary Of The Paper:**

This paper introduces the idea of diffusion-GAN by incorporating diffusion process with GANs. In the framework, an adaptive diffusion process for sampling the noise and a diffusion timestep-dependent discriminator are added to a traditional GAN generator. The main contributions are:
1. Propose a diffusion-GAN framework which combines the technique of diffussion models and GAN;
2. Theoretically and empirically show the effectiveness of the method and the ability of stabling GAN training;
3. According to the results reported, the quality of the generative images are better than the state-of-the-art GANs.

**Summary Of The Review:**

This paper studies a combination of two state-of-the-art generative models and theoretically shows its ability to stable the GAN training. According to the results, the images are of better quality than current GANs. I recommend accepting it.

---

> ### Author Response · Authors · 2022-11-11
> **Response**
>
> We thank reviewer YnnY for the positive feedback and helpful suggestions. We report the Inception Score (IS) for CIFAR-10 dataset below and also include this comparison in **Appendix K**. Note CIFAR-10 is a well-known dataset and tested by almost all baselines, so we picked CIFAR-10 and we referenced the reported IS values from their original papers for a fair comparison.
>
> |                         | IS   | FID  | Recall | NFE  |
> |-------------------------|------|------|--------|------|
> | Denoising Diffusion GAN | 9.63 | 3.75 | 0.57   | 4    |
> | DDPM                    | 9.46 | 3.21 | 0.57   | 1000 |
> | DDIM                    | 8.78 | 4.67 | 0.53   | 50   |
> | StyleGAN2               | 9.18 | 8.32 | 0.41   | 1    |
> | StyleGAN2 + DiffAug     | 9.40 | 5.79 | 0.42   | 1    |
> | StyleGAN2 + ADA         | 9.83 | **2.92** | 0.49   | 1    |
> | Diffusion StyleGAN2     | **9.94** | 3.19 | **0.58**   | 1    |

---

### Official Review · Reviewer_jaon · 2022-10-25

**Confidence:** 3
**Correctness:** 4
**Technical Novelty And Significance:** 3
**Empirical Novelty And Significance:** 4
**Recommendation:** 8

**Clarity, Quality, Novelty And Reproducibility:**

Overall the writting is good, the proposed method looks sound and in line with previously proposed methods.

**Strength And Weaknesses:**

Strength:
- The abstract and introduction give a nice overview and motivation for the problem.
- The method is supported by a theoretical analysis.
- The results are good and evaluation is reasonable.
- The objective is still differentiable wrt the generator using the classical rewritting of the perturbed samples.

Weaknesses:
- 25-gaussians example (Fig. 5): the purpose of the figure is to exhibit the fact that the propose method is not prone to mode collapse but I am curious to know if the proposed method has shown some mesurable improvement wrt SotA methods on this "toy example"?
- The authors did an ablation study on the adaptive diffusion process. But, reading Section 3.3, it feels like the adaptative diffusion strategy is key to the method whereas the ablation study does not clearly support it.

**Summary Of The Paper:**

The authors propose a diffusion-based approach to stabilise the learning process of GANs in a instance noise fashion.

**Summary Of The Review:**

Overall I think this is a good paper tackling a well motivated task.

---

> ### Author Response · Authors · 2022-11-11
> **Response**
>
> We thank reviewer jaon for the positive feedback and helpful suggestions. We provide clarifications below to answer your two questions.
>
> > 25-gaussians example (Fig. 5): the purpose of the figure is to exhibit the fact that the propose method is not prone to mode collapse but I am curious to know if the proposed method has shown some mesurable improvement wrt SotA methods on this "toy example"?
>
> This 25-gaussians toy example is mainly for the proof of concept that DiffusionGAN is not prone to mode collapse. We didn’t show quantified measurement of mode coverage here, since the mode coverage could be visualized. Note that, for the purpose of proof of concept, we built DiffusionGAN on a vanilla GAN backbone, which is MLP based and does not have any training tricks such as adding gradient penalty. We mentioned this in Section 4.2, “25-Gaussians Example” paragraph. Hence, in this experiment, DiffusionGAN is only compared with its GAN backbone visually, not with other methods. We further note that our practice of visually inspecting the mode coverage performance on a multimodal toy data is consistent with that in prior works, such as DDGAN in Xiao et al. (2021) paper.
>
> Xiao, Zhisheng, Karsten Kreis, and Arash Vahdat. "Tackling the Generative Learning Trilemma with Denoising Diffusion GANs." International Conference on Learning Representations. 2021.
>
>
> > The authors did an ablation study on the adaptive diffusion process. But, reading Section 3.3, it feels like the adaptative diffusion strategy is key to the method whereas the ablation study does not clearly support it.
>
> Thanks for the instructive suggestion. We have added one ablation study in Appendix I to analyze the adaptive diffusion strategy. As shown in Figure 7, we observe with the proposed adaptive diffusion strategy, the training curves of FIDs converge faster and reach lower final FIDs.

---

> > ### Comment · Reviewer_jaon · 2022-11-28
> > **Concluding answer**
> >
> > I thank the authors for the clarifications and extra information.
> > I consider my concerns addressed. I continue to favor acceptance.

---

### Author Response · Authors · 2022-11-11
**Response to All**

We thank all reviewers for the time and expertise they have invested in these reviews and for their positive and constructive feedback. We are encouraged that all reviewers praised the novelty and effectiveness of our method. Your comments and suggestions have helped us to improve the paper. We provide a response and clarifications below for each reviewer.

We have updated the paper accordingly,  with the changes highlighted blue:
* We have revised the details of hyper-parameters in Appendix G.
* We have added an ablation study of adaptive diffusion strategy in Appendix I.
* We have reported the Inception Score of Diffusion StyleGAN2 on CIFAR-10 in Appendix K.

---

### Decision · Program_Chairs · 2023-01-20

**Decision:**

Accept: poster

**Justification For Why Not Higher Score:**

Given the huge success of recent diffusion models in image generation, the significance of the proposed improved GAN training is questionable. Additionally, noise injection has been used in the past for GAN training.

**Justification For Why Not Lower Score:**

The paper is strong enough for publication at ICLR.

**Metareview: Summary, Strengths And Weaknesses:**

This paper improves GAN training by leveraging the forward diffusion process in diffusion models. The basic idea is that instead of only comparing real and generated data distribution directly using a discriminator, samples from both distributions are perturbed following the forward diffusion and they are contrasted to each using a time-dependent discriminator. This way the discriminator can provide feedback (more precisely data likelihood ratios) to the generator on all the intermediate smoother real/fake distributions. Theoretical analysis shows that matching real and fake distributions in the intermediate perturbed steps can in fact result in matching the base non-perturbed setting as well.

On the positive side, the paper is written clearly and motivations are stated nicely. The experimental results support the claims and the theoretical analysis provides insights into why the diffusion process can provide consistent training. Although prior works have considered Gaussian noise injection as an augmentation technique for training GANs, this paper provides new perspectives on this aspect. On minor issue with the proposed technique is that it does not fundamentally change the mode dropping behavior of GANs.

**Note From Pc:**

if the above contains the word "oral" or "spotlight" please see: "oral" presentation means -> notable-top-5% and "spotlight" means -> notable-top-25%. As stated in our emails, we are disassociating presentation type from AC recommendations

**Summary Of Ac-Reviewer Meeting:**

N/A